# The Value Function Semi-Algebraic Set in Partially Observable Markov Decision Processes

**Ryan A. Anderson** [* 1]   **Guido Montúfar** [* 1 2 3]

## Abstract

We study the geometry of feasible value functions in infinite-horizon partially observable Markov decision processes (POMDPs) under memoryless stochastic policies. Our main contribution is a characterization of the feasible set of value functions as a semi-algebraic set, defined by explicit polynomial inequalities determined by the transition dynamics, observation kernel, and reward structure of the POMDP. This result extends prior work for fully observable Markov decision processes, where the feasible set is known to be a polytope, to the substantially more intricate partially observable setting. In contrast to the polyhedral structure arising in MDPs, partial observability induces fundamentally nonlinear constraints, leading to a richer and more complex geometric structure. Our geometric characterization provides new insight into the landscape of policy optimization in both MDPs and POMDPs, and reveals qualitative phenomena unique to partial observability, including the emergence of isolated local maximizers of the long-term reward and their dependence on the initial state distribution.

## 1. Introduction

Markov decision processes (MDPs) provide a powerful mathematical framework for sequential decision-making problems and have been successfully deployed across a wide range of domains, including fisheries management (White, 1993), autonomous systems and drone warfare (Zhang et al., 2023), and game playing at superhuman levels (Silver et al., 2017). Originally introduced in the context of dynamic pro-

gramming for optimal control (Bellman, 1957), MDPs also form the foundation of modern reinforcement learning (Sutton & Barto, 2018).

While classical MDPs model uncertainty through stochastic state transitions, many real-world decision problems involve an additional layer of uncertainty arising from partial observability: the agent does not have direct access to the true system state and must act based solely on observations. This setting was formalized in the framework of partially observable Markov decision processes (POMDPs), beginning with the work of Åström (1965), followed by developments for the finite-horizon case by Smallwood & Sondik (1973) and for infinite-horizon discounted problems by Sondik (1978). Owing to their greater modeling flexibility, POMDPs have found numerous applications (Cassandra, 1998), particularly in robotics (Lauri et al., 2023), as well as in domains as diverse as elevator control (Crites & Barto, 1998) and the conservation of rare species (Chadès et al., 2021).

A growing body of work on the sample complexity of reinforcement learning in POMDPs has demonstrated that partial observability substantially increases the difficulty of learning optimal policies. One key challenge is that the observation process in a POMDP induces non-Markovian dynamics from the agent's perspective (Chen et al., 2023). For finite-horizon POMDPs, finding an optimal policy is exponential in the horizon length, implying that learning algorithms may require exhaustive exploration of the state space (Krishnamurthy et al., 2016). For discounted-reward, infinite-horizon POMDPs, the problem is undecidable in general (Papadimitriou & Tsitsiklis, 1987; Madani et al., 2003). This stands in sharp contrast to the fully observable case, where learning admits polynomial sample complexity in the horizon length (Azar et al., 2017). More recent work has identified structural conditions under which learning in POMDPs becomes tractable, such as $\alpha$-weakly revealing POMDPs, which admit polynomial sample complexity guarantees (Liu et al., 2022).

As in the fully observable setting, one may further distinguish between POMDPs that allow policies with memory and those restricted to be memoryless, depending only on the current observation. If the agent is permitted to maintain a belief state (an internal estimate of the latent state

---

[1]Department of Statistics & Data Science, University of California, Los Angeles, USA [2]Department of Mathematics, University of California, Los Angeles, USA [3]Max Planck Institute for Mathematics in the Sciences, Leipzig, Germany. Correspondence to: Ryan A. Anderson <raanderson@g.ucla.edu>.

*Proceedings of the 43$^{rd}$ International Conference on Machine Learning*, Seoul, South Korea. PMLR 306, 2026. Copyright 2026 by the author(s).

constructed from past observations) the POMDP can be reduced to a fully observable MDP (Åström, 1965; Subramanian et al., 2022). Although memoryless policies have historically been viewed as inadequate for POMDPs (Cohen & Parmentier, 2023), recent work shows that, when external memory is incorporated into the environment, memoryless policies can nonetheless achieve optimality (Toro Icarte et al., 2023).

In this work, we focus on the value function, a fundamental object in reinforcement learning that plays a central role in the characterization and computation of optimal policies. In fully observable Markov decision processes, standard algorithms such as value iteration are guaranteed to recover an optimal policy by acting greedily with respect to the current value function (Puterman, 2005).

Understanding the geometry of the set of feasible value functions provides insight into the structure and complexity of policy optimization in sequential decision-making problems. While recent work has developed geometric characterizations for MDPs, the corresponding theory for partially observable Markov decision processes has remained largely unexplored. This work aims to fill that gap.

## 1.1. Contributions

Our results can be summarized as follows:

- We interpret the set of feasible value functions of a (PO)MDP as the solution set of a parametric system, the Bellman equation parametrized by the policy, which allows us to obtain explicit descriptions (Section 3).

- We first provide an exact description in terms of infinitely many piecewise linear inequalities that serve as a sufficient and necessary condition for a given value function to be feasible (Theorem 3.9).

- We then obtain a description of the feasible set of value functions in terms of only finitely many polynomial equations and inequalities that constitute an explicit semi-algebraic description (Theorem 4.2).

- Finally, we provide an explicit construction of the feasible set of value functions, identifying a finite set of determinantal conditions on the parameters of the POMDP which form the boundary of the feasible set. (Theorem 4.4)

These results apply to both fully observable and partially observable settings, and highlight the role of the transition probabilities, observation probabilities, and instantaneous rewards on the structure of the value functions, and, in turn, the complexity of the policy optimization problem.

## 1.2. Related Works

For fully observable Markov decision processes, Dadashi et al. (2019) showed that the set of value functions is a union of convex polytopes. In particular, this set is closed and bounded, and can be described as a finite union of intersections of halfspaces. This geometric characterization has been leveraged to design more efficient algorithms for computing optimal policies, such as geometric policy iteration (Wu & De Loera, 2022). Moreover, Dadashi et al. (2019) established a *line theorem*, showing that when policies are fixed on all but one state, the corresponding value functions trace out a line segment. This result provides geometric intuition for the effectiveness of policy improvement and value iteration in fully observable MDPs.

In contrast, policy improvement is in general not feasible in partially observable settings, as distinct states may induce identical observations, introducing intrinsic coupling between decision variables (Li et al., 2011). As a result, the geometric structure underlying value functions in POMDPs is fundamentally different and less well understood.

Related geometric perspectives have been developed for variants of MDPs. For robust MDPs, in which transition kernels are selected adversarially at each round, the set of value functions has been shown to arise as the intersection of conic hypersurfaces (Wang et al., 2022).

We highlight the work of Müller & Montúfar (2022b), which showed that for memoryless, discounted, infinite-horizon (PO)MDPs, the value function and related quantities, such as state–action occupancy measures, can be expressed as rational functions of the policy parameters. They derived bounds on the degree of these rational parametrization in terms of the number of states consistent with a given observation, a quantity they term the degree of observability, and obtained a semi-algebraic description of the feasible set of state-action occupancy measures. These ideas have subsequently been used to develop optimization procedures (Müller & Montúfar, 2022a) and have been further studied in the context of state-aggregation (Dressler et al., 2024).

Despite these advances, the global geometric structure of the set of feasible value functions in partially observable Markov decision processes has, to our knowledge, remained unexplored.

## 2. Preliminaries

### 2.1. Markov Decision Processes

For a finite set $V$, we denote the probability simplex over $V$ as $\Delta_V = \{p \in \mathbb{R}^V : \sum_{v \in V} p(v) = 1, p(v) \geq 0\}$. For a pair of sets $V, W$, we denote the set of Markov kernels from $W$ to $V$ by $\Delta_V^W = \{p \in \mathbb{R}^{W \times V} : p(\cdot|w) \in \Delta_V, w \in W\}$. We sometimes write $p(w; v)$ for $p(v|w)$, where the first

argument indexes the rows of the kernel.

A Markov decision process (MDP) is a tuple $\mathcal{M} = \langle \mathcal{S}, \mathcal{A}, \mathcal{O}, \alpha, \beta, r, \gamma \rangle$, with states $\mathcal{S}$, actions $\mathcal{A}$, observations $\mathcal{O}$, transition kernel $\alpha : \mathcal{S} \times \mathcal{A} \to \Delta_{\mathcal{S}}$, observation kernel $\beta : \mathcal{S} \to \Delta_{\mathcal{O}}$, instantaneous reward function $r : \mathcal{S} \times \mathcal{A} \to \mathbb{R}$, and discount factor $\gamma$.

When $\beta$ is deterministic and injective, then the MDP is *fully observable*; otherwise $\mathcal{M}$ is a partially observable Markov decision process (POMDP).

Policies $\pi$ are kernels $\pi \colon \mathcal{O} \to \Delta_{\mathcal{A}}$, $\pi(a|o) = \mathbb{P}(\text{taking action } a|\text{observed } o)$. The observation kernel $\beta$ determines the probability of observing $o$ given the agent is in state $s$, $\beta(o|s) = \mathbb{P}(\text{observing } o|\text{agent in } s)$, while the transition kernel gives the next-state probability conditional upon being in state $s$ and taking action $a$, that is, $\alpha(s'|s, a) = \mathbb{P}(\text{next state is } s'|\text{in state } s, \text{taking action } a)$.

For any policy over observations $\pi \in \Delta_{\mathcal{A}}^{\mathcal{O}}$ we can define its effective policy over states. Concretely, for a fixed observation kernel $\beta$ we consider the map $\Delta_{\mathcal{A}}^{\mathcal{O}} \to \Delta_{\mathcal{A}}^{\mathcal{S}}$ given by $\tau = \pi \circ \beta$. Note this is a linear map as $\tau(a|s) = \sum_{o \in \mathcal{O}} \pi(a|o)\beta(o|s)$.

Given a policy, we further define the policy-weighted transition kernel $P^{\pi}(s'|s) \in \Delta_{\mathcal{S}}^{\mathcal{S}}$ via

$$P^{\pi}(s'|s) = \sum_{a \in \mathcal{A}} (\pi \circ \beta)(a|s)\alpha(s'|s, a) \qquad (1)$$

as well as a reward vector $r^{\pi} \in \mathbb{R}^{\mathcal{S}}$ via

$$r^{\pi}(s) = \sum_{a \in \mathcal{A}} (\pi \circ \beta)(a|s)r(s, a). \qquad (2)$$

For every policy $\pi$, we can calculate the value of the policy in a state $s$, which is the expected discounted sum of rewards obtained from beginning in state $s$ and following the policy:

$$V^{\pi}(s) = \mathbb{E}_{P^{\pi}}\left[\sum_{t=0}^{\infty} \gamma^t r(s_t, a_t)\bigg| s_0 = s\right].$$

The *value function* of the policy $\pi$ is obtained by taking the value of the policy for each state and assembling them into a vector $V^{\pi} \in \mathbb{R}^{\mathcal{S}}$. Note that we can also condition on the initial action to obtain the $Q$-value function (Sutton & Barto, 2018):

$$Q^{\pi}(s, a) = \mathbb{E}_{P^{\pi}}\left[\sum_{t=0}^{\infty} \gamma^t r(s_t, a_t)\bigg| s_0 = s, a_0 = a\right].$$

Bellman's optimality equation (Bellman, 1957) relates the value function of a policy to the transition kernel and reward vector it defines as follows:

$$(I - \gamma P^{\pi})V^{\pi} = r^{\pi}. \qquad (3)$$

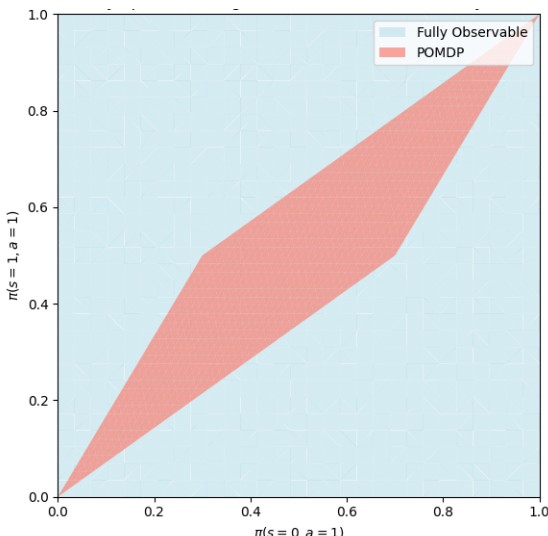

*Figure 1.* The effective policy polytope for fully observable 2-state MDPs is the blue square, while the red parallelogram is the effective policy polytope for a partially observable 2-state MDPs, for some $\beta$ not equal to the identity matrix.

For $\gamma \in (0, 1)$, $I - \gamma P^{\pi}$ is invertible and the Bellman equation has a unique solution for all $\pi \in \Delta_{\mathcal{A}}^{\mathcal{O}}$.

We regard the Bellman equation (3) as a *parametric linear system* in the indeterminate $V^{\pi} \in \mathbb{R}^{\mathcal{S}}$ whose coefficients $(I - \gamma P^{\pi}) \in \mathbb{R}^{\mathcal{S} \times \mathcal{S}}$ and $r^{\pi} \in \mathbb{R}^{\mathcal{S}}$ are parametrized by the policy $\pi$. From (1) and (2) we see that the parametrization is linear in the entries of the policy.

We are interested in the geometry of the set of solutions, i.e., the set of value functions, as the policy ranges over the set of policies over observations, that is, $\mathcal{V} = \{V^{\pi} : \pi \in \Delta_{\mathcal{A}}^{\mathcal{O}}\}$. The set $\mathcal{V}$ is the *solution set* to the Bellman equation as a parametric system with parameters $\pi \in \Delta_{\mathcal{A}}^{\mathcal{O}}$.

Note that for any $\pi \in \Delta_{\mathcal{A}}^{\mathcal{O}}$, the value function $V^{\pi}$ is given by the value function of its effective policy over states, $\tau = \pi \circ \beta \in \Delta_{\mathcal{A}}^{\mathcal{S}}$. Thus we may equivalently regard the system as being parametrized in terms of the effective policies, which are subject to corresponding constraints.

The set of effective policies $\tau = \pi \circ \beta \in \Delta_{\mathcal{A}}^{\mathcal{S}}$ for fixed observation kernel $\beta$ and arbitrary observation policy $\pi \in \Delta_{\mathcal{A}}^{\mathcal{O}}$ is the *effective policy polytope*, denoted $\Delta_{\mathcal{S}, \beta}^{\mathcal{A}}$. Note that this is indeed a polytope since it is the image of the polytope $\Delta_{\mathcal{A}}^{\mathcal{O}}$ under the linear map $\pi \to \pi \circ \beta$. We visualize the difference in the effective policy polytope between fully observable and partially observable 2-state MDPs in Figure 1.

## 2.2. Geometry of Value Functions in MDPs

Dadashi et al. (2019, Theorem 1) show that fixing the behavior of a policy $\pi$ on $k$ states amounts to constraining the value function into an $(|\mathcal{S}| - k)$-dimensional affine space.

This insight leads directly to the *line theorem*, which shows that as the policy varies at a single state $s$, the set of attainable value functions forms a one-dimensional line segment.

For a fully observable MDP, the set of value functions can be characterized using this approach as a union of polytopes.

**Theorem 2.1** (Dadashi et al., 2019, Theorem 3). *Let $\pi$ be a policy in a fully observable MDP. Consider states $s_1, \ldots, s_k$ and the set $Y^{\pi}_{s_1,\ldots,s_k}$ of all policies that agree with $\pi$ on $s_1, \ldots, s_k$. Then the image of $Y^{\pi}_{s_1,\ldots,s_k}$ under the map $(I - \gamma P^{\pi})^{-1} r^{\pi}$ is a (non-convex) polytope. Moreover, the image of $Y^{\pi}_{\emptyset}$ is a (non-convex) polytope.*

## 3. Feasible Value Functions as a Solution Set

We introduce interval and parametric linear systems and review key results in this context.

Later we will explain how the results on MDPs in Section 2.2 can be derived using the framework of interval systems. Then we will use the framework of parametric systems to generalize the results to the case of POMDPs.

### 3.1. Interval and Parametric Linear Systems

Interval linear systems are linear systems $Ax - b = 0$ where the matrix $A \in \mathbb{R}^{m \times n}$ and the vector $b \in \mathbb{R}^m$ have entries specified only up to intervals.

Such systems have been studied in the literature since the 1960s. In particular, the solution sets of interval linear systems are characterized in the Oettli-Prager theorem that we discuss below.

We write $[A]$ for a matrix of intervals, i.e., a set of matrices $A$ whose entries satisfy $A_{ij} \in [\underline{a_{ij}}, \overline{a_{ij}}]$, for some $\underline{a_{ij}}, \overline{a_{ij}} \in \mathbb{R}$.

Given such an interval matrix, we write $A^c$ for the corresponding matrix of interval centers, i.e., the matrix with entries $A^c_{ij} = \frac{1}{2}(\underline{a_{ij}} + \overline{a_{ij}})$, and $A^{\Delta}$ for the corresponding matrix of interval lengths, i.e., the matrix with entries $(A^{\Delta})_{ij} = \frac{1}{2}(\overline{a_{ij}} - \underline{a_{ij}})$.

**Theorem 3.1** (Oettli & Prager, 1964, as stated by Rohn, 1989, eq. 1). *A vector $x \in \mathbb{R}^n$ is the solution to $Ax - b = 0$ for some $m \times n$-matrix $A \in [A]$ and some $m$-vector $b \in [b]$ if and only if*

$$|A^c x - b^c| \leq A^{\Delta}|x| + b^{\Delta}.$$

*Remark* 3.2. The condition stated in Theorem 3.1 defines a set $S = \cup_{r \in \{+1,-1\}^n} P_r$, where $P_r$ is the subset of $\mathbb{R}^n$ cut by the $n$ linear inequalities that define the $r$-th orthant, $\mathrm{diag}(r)x \geq 0$, and $2m$ further linear inequalities $A^{\Delta} \mathrm{diag}(r)x + b^{\Delta} \pm (A^c x - b^c) \geq 0$. Of the $n + 2m$ inequalities that define each $P_r$ only $2m$ are boundaries of $S$ (see, e.g., Alefeld et al., 2001, Theorem 1, f).

Now let us discuss a generalization from interval to parametric linear system $A(p)x - b(p)$, where $A(p)$ and $b(p)$ depend linearly on a parameter $p$. We consider an affine linear parametrization of the form

$$A(p) = A^0 + \sum_{k=1}^{K} A^k p_k, \quad b(p) = b^0 + \sum_{k=1}^{K} b^k p_k, \quad (4)$$

with fixed $A^k$ and $b^k$, and a parameter vector $p$ with entries satisfying $p_k \in [\underline{p_k}, \overline{p_k}]$.

We can convert any interval system into a parametric system by allowing each entry of the parametric matrix to vary as its own parameter (Popova, 2012). By contrast, moving from a parametric system to an interval system is more difficult in general, if at all possible. For parametric systems, the direct analog of the Oettli-Prager theorem only provides a necessary but not sufficient condition for a given vector $x$ to solve a system $A(p)x - b(p) = 0$.

**Theorem 3.3** (Hladík, 2012, Theorem 2). *Let $[p] = \times_{k=1}^K [\underline{p_k}, \overline{p_k}]$, $p_k^c = \frac{1}{2}(\overline{p_k} + \underline{p_k})$ and $p_k^{\Delta} = \frac{1}{2}(\overline{p_k} - \underline{p_k})$. Let $A(p) = A^0 + \sum_{k=1}^K p_k A^k$, $b(p) = b^0 + \sum_{k=1}^K p_k b^k$ for $p \in [p]$. If a vector $x \in \mathbb{R}^n$ is the solution to $A(p)x - b(p) = 0$ for some $p \in [p]$, then*

$$|A(p^c)x - b(p^c)| \leq \sum_{k=1}^{K} p_k^{\Delta}|A^k x - b^k|.$$

We refer to this necessary condition as an *enclosure for the solution set*. The result can be interpreted as a finite list of piecewise linear inequalities which enclose the solution set. *Remark* 3.4. The necessary condition in Theorem 3.3 defines a set of the form

$$S = \cap_{i=1}^m S_i, \quad S_i = \cup_{r \in \{+1,-1\}^{K+1}} P_{i,r},$$

where each $P_{i,r}$ is a polyhedron defined by $K + 2$ linear inequalities in $\mathbb{R}^n$, only one of which is a boundary of $S_i$. Note that some of the $P_{i,r}$ may be empty and thus redundant. Equivalently, distributing the intersections over the unions gives a representation as a union of $2^{m(K+1)}$ polyhedra of the form $\cap_{i=1}^m P_{i,r^i}$, each defined by $m(K + 2)$ linear inequalities, of which only $m$ are boundaries of $S$.

Hladík (2012) also provides a condition for parametric matrix systems which is both necessary and sufficient for defining the solution set.

**Theorem 3.5** (Hladík, 2012, Theorem 3). *Let $[p] = \times_{k=1}^K [\underline{p_k}, \overline{p_k}]$, $p_k^c = \frac{1}{2}(\overline{p_k} + \underline{p_k})$ and $p_k^{\Delta} = \frac{1}{2}(\overline{p_k} - \underline{p_k})$. Let $A(p) = A^0 + \sum_{k=1}^K p_k A^k$, $b(p) = b^0 + \sum_{k=1}^K p_k b^k$ for $p \in [p]$. Then $x \in \{x \in \mathbb{R}^n : A(p)x = b(p) \text{ for some } p \in [p]\}$ if and only if for every $y \in \mathbb{R}^n$ it solves*

$$y^{\top}(A(p^c)x - b(p^c)) \leq \sum_{k=1}^{K} p_k^{\Delta}|y^{\top}(A^k x - b^k)|.$$

*Remark* 3.6. Note that this description involves infinitely many piecewise linear inequalities, one for each choice of $y$. Each inequality verifies the non-negativity of a continuous piecewise linear function with up to $2^K$ linear pieces.

### 3.2. Parametric System for the Bellman Equation

Now we provide an explicit description of the Bellman equation as a parametric linear system with an affine linear parameterization in the policy.

The system (3) with the transition kernel given in (1) and the reward vector given in (2) can be written as $A(p)x - b(p) = 0$, where the coefficients parametrization $(A(p), b(p)) \in \mathbb{R}^{\mathcal{S} \times \mathcal{S}} \times \mathbb{R}^{\mathcal{S}}$ is given by

$$A(p) = A^0 + \sum_{(o,a) \in \mathcal{O} \times \mathcal{A}} A^{(o,a)} p_{o,a}, \quad \text{with} \quad (5)$$

$$(A^0)_{s,s'} = I_{s,s'},$$
$$(A^{(o,a)})_{s,s'} = -\gamma \alpha(s,a;s')\beta(s;o), \quad (o,a) \in \mathcal{O} \times \mathcal{A},$$

and

$$b(p) = b^0 + \sum_{(o,a) \in \mathcal{O} \times \mathcal{A}} b^{(o,a)} p_{o,a}, \quad \text{with} \quad (6)$$

$$(b^0)_s = 0,$$
$$(b^{(o,a)})_s = r(s,a)\beta(s;o), \quad (o,a) \in \mathcal{O} \times \mathcal{A},$$

with parameter $p = (p_{o,a}) \in \Delta_{\mathcal{A}}^{\mathcal{O}} \subseteq \mathbb{R}^{\mathcal{O} \times \mathcal{A}}$.

We denote by $L$ the map $p \mapsto (A(p), b(p))$ defined in (5)–(6) and write $S = L(\Delta_{\mathcal{A}}^{\mathcal{O}})$ for the image of the policy polytope.

We would like to leverage the result of Hladík (2012) to characterize the solution set to the above system. However, the result given in Theorem 3.5 requires an affine parametrization by a hyperrectangle. To address this problem, we will represent $S$ as an intersection of sets parametrized by hyperrectangles. Then we can describe the solution set in terms of the combined set of inequalities given by Theorem 3.5 for each of the corresponding parametric systems.

**Intersection of Rectangular Parametrizations**  For each $i = 1, \ldots, |\mathcal{A}|$, fix $\mathcal{A}_i = \mathcal{A} \setminus \{a_i\}$ and define the parametrization $(B_i(v), c_i(v)) \in \mathbb{R}^{\mathcal{S} \times \mathcal{S}} \times \mathbb{R}^{\mathcal{S}}$ by

$$B_i(v) = B_i^0 + \sum_{(o,a) \in \mathcal{O} \times \mathcal{A}_i} B_i^{(o,a)} v_{o,a}, \quad \text{with} \quad (7)$$

$$(B_i^0)_{s,s'} = (A^0)_{s,s'} + \sum_{o \in \mathcal{O}} (A^{(o,a_i)})_{s,s'},$$
$$(B_i^{(o,a)})_{s,s'} = (A^{(o,a)})_{s,s'} - (A^{(o,a_i)})_{s,s'}, \quad (o,a) \in \mathcal{O} \times \mathcal{A}_i,$$

and

$$c_i(v) = c_i^0 + \sum_{(o,a) \in \mathcal{O} \times \mathcal{A}_i} c_i^{(o,a)} v_{o,a}, \quad \text{with} \quad (8)$$

$$(c_i^0)_s = \sum_{o \in \mathcal{O}} (b^{(o,a_i)})_s,$$
$$(c_i^{(o,a)})_s = (b^{(o,a)})_s - (b^{(o,a_i)})_s, \quad (o,a) \in \mathcal{O} \times \mathcal{A}_i,$$

with parameter $v \in [0,1]^{\mathcal{O} \times \mathcal{A}_i}$. We write $v^c$ for the center of the parameter region, which is the matrix with entries $(v^c)_{o,a} = \frac{1}{2}$. Similarly, we write $v^\Delta$ for the matrix of possible deviations from the center, $(v^\Delta)_{o,a} = \frac{1}{2}$.

Denote by $S_i$ the image of the parametrization (7)–(8). Note that $S_i = L(E_i)$, where $E_i$ is the image of the map $[0,1]^{\mathcal{O} \times \mathcal{A}_i} \to \mathbb{R}^{\mathcal{O} \times \mathcal{A}}$, $v \mapsto p$ given by $p_{o,a_i} = 1 - \sum_{a \neq a_i} v_{o,a}$, $p_{o,a} = v_{o,a}$, $a \neq a_i$.

**Proposition 3.7.** *Assume that $L(E_i) \cap L(E_j) = L(E_i \cap E_j)$ for all $i, j$, or, more strongly, that $L$ is injective on $\bigcup_i E_i$. Then $S = \bigcap_i S_i$.*

*Proof.* To be found in Appendix A. $\qquad\square$

*Remark* 3.8. We observe that if $|\mathcal{O}| \leq |\mathcal{S}|$ and $|\mathcal{A}| \leq |\mathcal{S}|+1$, then the assumption in Proposition 3.7 is satisfied for generic choices of $\alpha, \beta, r$, with details in Appendix A.

### 3.3. Solution Set of the Bellman Equation

With the description of Proposition 3.7 at hand, we can now leverage Theorem 3.5 to characterize the solution set of the Bellman equation parametrized by the policy polytope.

**Theorem 3.9** (Value functions of POMDPs). *Consider a (PO)MDP. Suppose the assumption of Proposition 3.7 holds. Then $x \in \mathbb{R}^{\mathcal{S}}$ is a feasible value function, meaning that it solves the Bellman equation $(I - \gamma P^\pi)x - r^\pi = 0$ for some $\pi \in \Delta_{\mathcal{A}}^{\mathcal{O}}$ if and only if for every $y \in \mathbb{R}^n$ and every $i \in \{1, \ldots, |\mathcal{A}|\}$ it satisfies*

$$y^\top (B_i(v^c)x - c_i(v^c)) \leq$$
$$\sum_{(o,a) \in \mathcal{O} \times \mathcal{A}_i} v_{o,a}^\Delta |y^\top (B_i^{(o,a)}x - c_i^{(o,a)})|,$$

*where the matrices are given in eqs. 5–6 and 7–8.*

*Proof.* This follows from the description of the Bellman system given in Section 3.2, particularly Proposition 3.7, and the characterization of solution sets given in Theorem 3.5. $\qquad\square$

In Figure 2 we illustrate the characterization provided by Theorem 3.9, in which the feasible set of value functions for a POMDP is described via infinitely many piecewise linear inequalities.

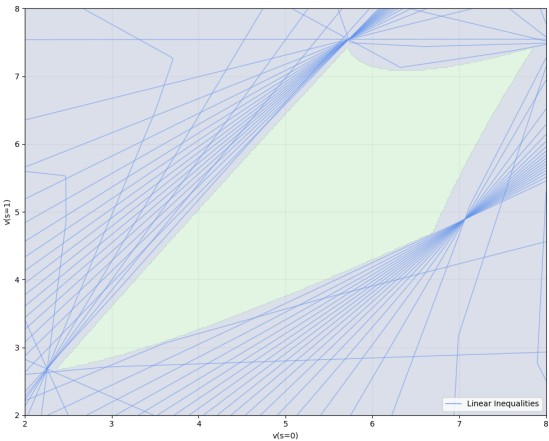

*Figure 2.* With Theorem 3.9, we are able to identify the feasible space of value functions for any POMDP as an intersection of infinitely many piecewise linear inequalities. Here we sampled 50 such inequalities for a POMDP as light blue lines and show they enclose the feasible region, shown in green.

*Remark* 3.10. In the special case of fully observable MDPs, Theorem 3.9 reduces to finitely many piecewise linear inequalities. For details, see Appendix B.

We can obtain an enclosure analog to Theorem 3.3 for the feasible value functions of POMDPs as a special case of Theorem 3.9.

**Corollary 3.11** (Enclosure of Value Functions of POMDPs). *Consider a POMDP. Then the set of feasible value functions, i.e., solutions to the Bellman equation $(I - \gamma P^\pi)x - r^\pi = 0$, is contained within the set defined by the following constraints:*

$$|B_i(v^c)x - c_i(v^c)| \leq$$
$$\sum_{(o,a)\in\mathcal{O}\times\mathcal{A}_i} v_{o,a}^\Delta |(B_i^{(o,a)}x - c_i^{(o,a)})|.$$

*Proof.* This follows from Theorem 3.9 by setting $y$ equal to the sign of each component of $B_i(v^c)x - c_i(v^c)$ and noting that the inclusion $L(E_i) \cap L(E_j) \supseteq L(E_i \cap E_j)$ always holds. In that case, we have

$$y^\top(B_i(v^c)x - c_i(v^c)) =$$
$$(\text{sign}(B_i(v^c)x - c_i(v^c)))^\top(B_i(v^c)x - c_i(v^c))$$
$$= |(B_i(v^c)x - c_i(v^c))|$$

and

$$|y^\top(B_i^{(o,a)}x - c_i^{(o,a)})| =$$
$$|(\text{sign}(B_i(v^c)x - c_i(v^c)))^\top(B_i^{(o,a)}x - c_i^{(o,a)})|$$
$$= |(B_i^{(o,a)}x - c_i^{(o,a)})|$$

and the desired relation follows. □

# 4. Finite Description of the Solution Set

Theorem 3.9 gives a necessary and sufficient condition for a vector to be a feasible value function of a POMDP. However, it requires infinitely many piecewise linear inequalities to do so. In this section we seek an alternative and equivalent characterization that requires only finitely many equations and inequalities. We shall refer to such a characterization as a *finite description* of a solution set.

## 4.1. Finite Description for Parametric Systems

We first write the following criterion that we will then use to extract a finite description.

**Proposition 4.1.** *A vector $x$ solves a parametric matrix system $A(p)x - b(p) = 0$ with parametrization (4) if and only if there exist $q_k \in [-1, 1]$, $k = 1, \ldots, K$ such that*

$$A(p^c)x - b(p^c) = \sum_{k=1}^{K} q_k p_k^\Delta (A^k x - b^k).$$

*Proof.* To be found in Appendix C. □

Let $D$ be the matrix of deviations that appear on the right hand side of the statement, with the $k$-th column of $D$ given as

$$D_k(x) = p_k^\Delta (A^k x - b^k).$$

Similarly, let $C$ be the vector of midpoint residuals that appear on the left hand side,

$$C(x) = A(p^c)x - b(p^c).$$

Then we can rephrase the necessary and sufficient condition stated in Proposition 4.1 in terms of these matrices as

$$C = Dq, \quad \text{for some } -1 \leq q_k \leq 1. \quad (9)$$

This means that $C(x)$ is contained in the zonotope generated by the columns of $D(x)$. We interpret this as two conditions:

1. Equations: $C(x)$ is contained in $\text{col}(D(x))$.

2. Inequalities: the projection of $C(x)$ onto $\text{col}(D(x))$ satisfies the facet-defining inequalities of the zonotope generated by the columns of $D(x)$.

**Equality Condition** The first condition is equivalent to asking that all $(r + 1) \times (r + 1)$ sub-matrices of $[D|C]$ have determinant zero, where $r = \text{rank}(D)$. This gives us a system of polynomial equations in $x$ of degree $r + 1$. Note that the particular equations that need to be verified depend on $r = \text{rank}(D(x))$ and thus on $x$ itself. This is not a fundamental issue, as we can stratify the space of candidate value functions by the rank.

It is worth noting an equivalent condition that does not explicitly involve $\mathrm{rank}(D)$. Namely, the condition is equivalent to requiring that the projection of $C$ onto $\mathrm{col}(D)^\perp$ vanishes. Recall that, for any matrix $A \in \mathbb{R}^{m \times n}$ and vector $b \in \mathbb{R}^n$, the component of $b$ orthogonal to the column space of $A$ is given by $(I - AA^+)b$, where $A^+$ denotes the Moore-Penrose pseudoinverse of $A$ (see, e.g., Axler, 1997, Theorem 6.69).

In our setup, this condition becomes:

$$C^\perp = (I - DD^+)C = 0. \tag{10}$$

At those $x$ where $D$ has full column rank or full row rank, the Moore-Penrose pseudo inverse $D^+$ can be written explicitly as $(D^\top D)^{-1}D^\top$ and $D^\top(DD^\top)^{-1}$, respectively. In these cases, using $A^{-1} = \det(A)^{-1}\,\mathrm{adj}(A)$, we can write (10) as a system of polynomial equations in $x$.

**Inequality Condition**  To elaborate the second condition, recall that a zonotope $Z(g_1, \ldots, g_K)$ generated by vectors $g_1, \ldots, g_K$ is

$$Z(g_1, \ldots, g_K) = \Big\{ \sum_{k=1}^K \alpha_k g_k : \alpha_k \in [-1, 1] \Big\}.$$

A vector $h$ lies within the zonotope $Z(g_1, \ldots, g_K)$ if and only if we can verify via the facet-defining inequalities that $w^\top h \leq \sum_k |w^\top g_k|$, where $w$ is orthogonal to a maximal set of linearly independent generators that span said facet (see Proposition C.1). Thus in the case of our parametric systems we have inequalities of the form

$$w^\top(DD^+C) \leq \sum_k |w^\top D_k|, \tag{11}$$

where $w$ is any column of $(I - D_I D_I^+)$, i.e., a vector orthogonal to the columns of $D_I$, and $D_I$ is a sub-matrix of $D$ collecting linearly independent columns.

Again, at those points $x$ where $D(x)$ has full column rank or full row rank, we can write $D^+$ explicitly and (11) is a polynomial inequality in $x$.

### 4.2. Finite Description for the Bellman Equation

Finally, we rewrite the necessary and sufficient condition on feasible value functions given by Theorem 3.9 as a finite description by imposing a zonotope condition on the deviation and midpoint residuals of the parametrized Bellman equation.

**Theorem 4.2** (Zonotope Characterization of the Solution Set to the Bellman Equation). *Consider the parametrization of a POMDP given in eqs. 5-6 and 7-8. This system has deviation matrices $D_i$ with columns*

$$D_{i,(o,a)} = v_{(o,a)}^\Delta(B_i^{(o,a)}x - c_i^{(o,a)})$$

*and midpoint residuals $C_i$, with entries*

$$C_i = B_i(v^c)x - c_i(v^c)$$

*where $v^c, v^\Delta$ are the centers and widths, respectively, of the parameters, all equal to $\frac{1}{2}$.*

*Then $x \in \mathbb{R}^\mathcal{S}$ is a feasible value function, meaning that it solves the Bellman equation $(I - \gamma P^\pi)x - r^\pi = 0$ for some $\pi \in \Delta_\mathcal{A}^\mathcal{O}$, if and only if there exist vectors $q^{(i)} \in \mathbb{R}^{\mathcal{O} \times \mathcal{A}_i}$ such that*

$$D_i q^{(i)} = C_i, \ -1 \leq q^{(i)} \leq 1.$$

*Proof.* We noted in Proposition 3.7 that the Bellman equation can be parametrized as described in eqs. 7–8. Since the solution set for each of these parametric systems is itself described by the result in Proposition 4.1, we can obtain our desired result by enforcing they all hold simultaneously. □

In Figure 3 we compare the characterizations given in Theorem 4.2 and Theorem 3.5, which describe the feasible region for a POMDP with infinitely many piecewise linear inequalities and with finitely many nonlinear inequalities, namely the solutions to $|q^{(i)}| = 1$. Additional details on how we created Figures 2 and 3 can be found in Appendix E.

**Corollary 4.3.** *For any POMDP, the solution set to the Bellman equation, which is the set of feasible value functions, is a semi-algebraic set.*

*Proof.* To be found in Appendix C. □

Theorem 4.2 presents a description of the solution set to the Bellman equation in terms of the existence of certain coefficients. Now we provide a finite description of the solution set in terms of a finite list of explicit polynomial equations and inequalities. This construction provides additional insight into the number of constraints needed to form the feasible set, as well as the degree of those constraints, which are polynomial in the value function.

**Theorem 4.4** (Semi-algebraic description of the solution set to the Bellman equation). *Consider a POMDP with finite state space $\mathcal{S}$, observation space $\mathcal{O}$, and action space $\mathcal{A}$. Fix orderings of $\mathcal{O}$ and $\mathcal{A}$, and write a stochastic memoryless policy as a vector $P \in \mathbb{R}^{|\mathcal{O}||\mathcal{A}|}$ with entries satisfying $P(a|o) \geq 0, \sum_{a \in \mathcal{A}} P(a|o) = 1$ for all $o \in \mathcal{O}$.*

*For each $(o, a) \in \mathcal{O} \times \mathcal{A}$, define $u_{o,a}(x) \in \mathbb{R}^{|\mathcal{S}|}$ by*

$$u_{o,a}(x)_s = \beta(s; o)\left(\gamma \sum_{s' \in \mathcal{S}} \alpha(s, a; s')x_{s'} + r(s, a)\right).$$

*Let $d(x) = x \in \mathbb{R}^{|\mathcal{S}|}$, and let $R \in \mathbb{R}^{|\mathcal{O}| \times |\mathcal{O}||\mathcal{A}|}$ be the matrix encoding the row-sum constraints, so that $(RP)_o = \sum_{a \in \mathcal{A}} P(a|o)$.*

*Define $M(x) \in \mathbb{R}^{|\mathcal{S}| \times |\mathcal{O}||\mathcal{A}|}$ as the matrix whose columns are $u_{o,a}(x)$, and set*

$$C(x) = \begin{pmatrix} M(x) \\ R \end{pmatrix}, \qquad f(x) = \begin{pmatrix} d(x) \\ \mathbf{1}_{|\mathcal{O}|} \end{pmatrix}.$$

*Then $x$ is a feasible value function if and only if there exists $P \in \mathbb{R}^{|\mathcal{O}||\mathcal{A}|}$ such that $C(x)P = f(x), P \geq 0$.*

*Moreover, the solution set $S$ admits the following quantifier-free semi-algebraic description as a finite union stratified by $\text{rank } C(x) = \rho$:*

$$S = \bigcup_{\rho=1}^{|\mathcal{S}|+|\mathcal{O}|} \bigcup_{\substack{I \subseteq \{1,\ldots,|\mathcal{S}|+|\mathcal{O}|\} \\ |I|=\rho}} \bigcup_{\substack{B \subseteq \{1,\ldots,|\mathcal{O}||\mathcal{A}|\} \\ |B|=\rho}} S_{\rho,I,B},$$

*where $S_{\rho,I,B}$ is the set of all $x \in \mathbb{R}^{\mathcal{S}}$ satisfying the following polynomial inequalities:*

*(i) $\det C_{I,B}(x) \neq 0$ and every $(\rho+1) \times (\rho+1)$ minor of $C(x)$ vanishes;*

*(ii) every $(\rho+1) \times (\rho+1)$ minor of the augmented matrix $[C(x)|f(x)]$ vanishes;*

*(iii) for all $t = 1, \ldots, \rho$, $\det C_{I,B,t}(x) \det C_{I,B}(x) \geq 0$.*

*Here, for index sets $I \subseteq \{1, \ldots, |\mathcal{S}| + |\mathcal{O}|\}$ and $B \subseteq \{1, \ldots, |\mathcal{O}||\mathcal{A}|\}$ with $|I| = |B| = \rho$, the matrix $C_{I,B}(x)$ denotes the $\rho \times \rho$ submatrix of $C(x)$ with row set $I$ and column set $B$. $C_{I,B,t}(x)$ is obtained from $C_{I,B}(x)$ by replacing its $t$-th column by $f_I(x)$, the corresponding subvector of $f(x)$.*

*Proof.* To be found in Appendix D. □

Theorem 4.4 provides a finite description of the solution set to the Bellman equation, meaning an explicit semi-algebraic description of the solution set in terms of finitely many polynomial equations and inequalities. An example of the polynomial conditions we obtain in this way is described in Appendix E.4.

*Remark* 4.5. Nonconvex polyhedra, which admit a description as finite unions of intersections of finitely many half-spaces, are also semi-algebraic sets. The polyhedral characterization of the solution set of the Bellman equation for fully observable MDPs provided by Dadashi et al. (2019) can be seen as a special case of our characterization for (PO)MDPs.

## 5. Implications and Examples

Given Theorem 4.2, the boundary of the feasible set of value functions can be computed exactly as the solution set of the

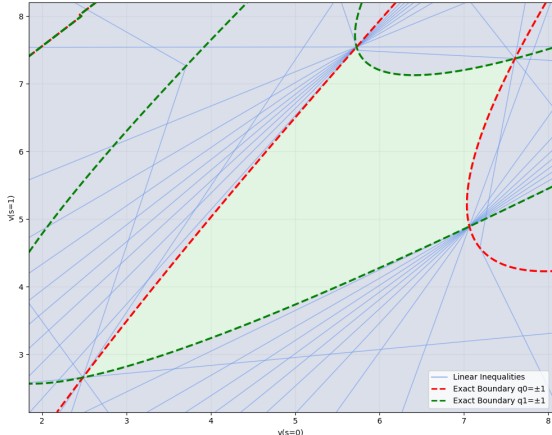

*Figure 3.* We are able to identify the feasible space of value functions for the POMDP with only four curves using Theorem 4.2, namely the solutions to $|q^{(1)}|, |q^{(2)}| = 1$, whose solutions are plotted as the red and green dashed curves. The faint blue lines are samples from the infinitely many piecewise linear inequalities needed to specify the feasible space under Theorem 3.9.

equation $Dq = C$. Since the objective of reinforcement learning is to identify an optimal policy, and optimal policies correspond to boundary points of the feasible value function set, understanding this boundary geometry provides insight into the structure of policy optimization problems.

**Linear Programming Fails to Reach Optima in POMDPs**
In fully observable MDPs, the optimal policy is independent of the initial state distribution $\rho$ and can be obtained by maximizing the linear objective $J = \sum_s \rho_s V^\pi(s)$ over the feasible set of value functions, for any choice of $\rho$.

Once a value function is optimal for one initial distribution, it is optimal for all choices of $\rho$ (Puterman, 2005, Section 5.9.1).

In contrast, no such invariance holds in partially observable MDPs. Figure 4 illustrates that in POMDPs, optimal policies need not be unique and may depend sensitively on the choice of the initial state distribution. As $\rho$ varies, both the identity and the multiplicity of optimal policies may change, reflecting the nonlinear geometry of the feasible set.

The geometric structure of this feasible set provides a natural lens through which to identify regions of initial state distributions that induce the same optimal policy, as well as to characterize the presence and number of isolated local maximizers of the long-term reward.

**Investigating Optimization Dynamics via Value and Policy Spread** To illustrate this point, we conducted a systematic study across randomly generated POMDPs. For each configuration (S, A, O), we sampled independent instances and ran 50 random restarts of policy gradient over memory-

less stochastic policies. Table 1 in Appendix F reports for various configurations aggregate statistics over instances, including the spread of achieved values across restarts, the fraction of materially suboptimal runs (gap > 0.01), and the spread of the resulting policies.

Across all configurations, we consistently observe that partial observability induces a substantial spread in both value and policy space, whereas the fully observable baseline exhibits negligible spread.

**Multiple Optima in the Optimization Landscape**  To more directly quantify the multi-optima structure, we ran another experiment where for each initial state distribution $\rho$, we clustered the converged value vectors. The number of clusters directly estimates the number of distinct local optima reached. Table 2 in Appendix F reveals that multiple distinct local optima are the norm rather than the exception. The fraction of (instance, $\rho$) pairs exhibiting multiple optima increases with $S$ as does the mean number of distinct local optima.

**Extension to Finite Memory Policies**  To further investigate whether the observed optimization landscape is an artifact of restricting to memoryless policies, we extended our experiments to include finite-memory policies implemented via observation enhancement.

Table 3 in Appendix F shows a consistent pattern across all tested configurations: Increasing memory reduces the value spread but partial observability still induces significant spread.

This is consistent with the semi-algebraic characterization of the feasible value set. Increasing $k$ enlarges the policy class and the set of feasible value functions, which can mitigate some optimization difficulties, reflected in the reduced spread. However, the geometric complexity induced by partial observability persists, as evidenced by the remaining variability compared to the fully observable case.

These observations are consistent with our theoretical results stating that, under partial observability, the set of achievable value functions induced by memoryless policies forms a semi-algebraic set with nontrivial geometry, including curved boundaries, which can give rise to multiple local optima of different values.

## 6. Conclusion

In this work, we developed a geometric characterization of the set of feasible value functions for infinite-horizon POMDPs under memoryless stochastic policies. By showing that this set admits an explicit semi-algebraic description, we extend classical polyhedral results for fully observable MDPs to the partially observable setting, where feasibility is

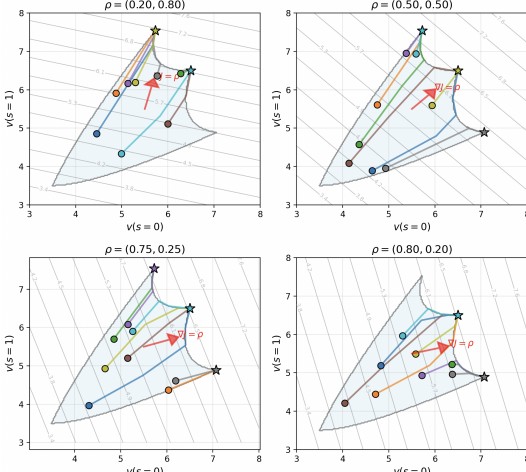

*Figure 4.* Above we sample eight feasible policies in the same POMDP with different choices of state distribution $\rho$ in each plot. Optimal policies, which differ between choices of $\rho$, are denoted with a star, while the red arrow corresponds to the steepest ascent direction for the objective. The grey dashed lines are level sets of the objective $J = c$. Here, $\rho$ represents an initial state distribution.

governed by fundamentally nonlinear constraints. This perspective clarifies qualitative differences between MDPs and POMDPs, including the dependence of optimal policies on the initial state distribution and the emergence of isolated local optima. More broadly, our results suggest that geometric structure plays a central role in understanding optimization and learning in partially observable environments.

Our analysis is restricted to memoryless stochastic policies and discounted infinite-horizon settings. While this restriction enables a tractable and explicit geometric characterization, extending the framework to policies with memory or belief-states remains an important open direction. A further limitation is that our results are primarily structural and do not directly translate into efficient algorithms for policy optimization in POMDPs; developing methods that exploit the identified geometry is therefore a natural next step. In particular, a more refined understanding of the semi-algebraic constraints and the development of computationally scalable approximations warrant further investigation. More broadly, elucidating how this semi-algebraic structure can inform algorithm design, convergence guarantees, and robustness analysis is a promising direction for future work.

## Acknowledgment

We would like to thank Johannes Müller for insightful discussions and valuable input during the initial development of this project. This project has been supported by NSF grant DMS-2522495. GM was partially supported by DARPA AIQ grant HR00112520014, NSF grants DMS-2145630, CCF-2212520, DFG SPP 2298 grant 464109215,

and BMFTR in DAAD project 57616814 (SECAI).

## Impact Statement

This work contributes to the mathematical foundations of machine learning through the study of the semi-algebraic geometry of value function sets in partially observable Markov decision processes. A deeper structural understanding of reinforcement learning systems may support future advances in optimization, verification, robustness, and interpretability of sequential decision-making models. The present work is theoretical in nature and does not involve deployment-oriented systems or human-subject data. We do not anticipate direct negative societal consequences stemming from the results presented here.

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

# Appendix Contents

This appendix is organized as follows:

# A. Proofs of Results in Section 3

*Remark* 3.4. The necessary condition in Theorem 3.3 defines a set of the form

$$S = \cap_{i=1}^m S_i, \quad S_i = \cup_{r \in \{+1,-1\}^{K+1}} P_{i,r},$$

where each $P_{i,r}$ is a polyhedron defined by $K + 2$ linear inequalities in $\mathbb{R}^n$, only one of which is a boundary of $S_i$. Note that some of the $P_{i,r}$ may be empty and thus redundant. Equivalently, distributing the intersections over the unions gives a representation as a union of $2^{m(K+1)}$ polyhedra of the form $\cap_{i=1}^m P_{i,r^i}$, each defined by $m(K + 2)$ linear inequalities, of which only $m$ are boundaries of $S$.

*Details on Remark 3.4.* To see this, note that the stated condition comprises $m$ inequalities $0 \le F_i(x) := \sum_k p_k^\Delta |A_{i:}^k x - b_i^k| - |A(p^c)_{i:} x - b(p^c)_i|$, $i = 1, \ldots, m$. Each of these inequalities verifies the non-negativity of a continuous piecewise linear function $F_i$. This function has one linear piece for each possible sign of the terms inside the absolute values, $r \in \{+1, -1\}^{K+1}$. Its solution set is the union of the solution sets over all linear pieces, $S_i = \cup_r P_{i,r}$. Each $P_{i,r}$ is a polyhedron defined by $K + 1$ linear inequalities that determine the particular linear region of $F_i$,

$$r_1(A_{i:}^1 x - b_i^1) \ge 0$$

$$\vdots$$

$$r_K(A_{i:}^K x - b_i^K) \ge 0$$

$$r_{K+1}(A(p^c)_{i:} x - b(p^c)_i) \ge 0,$$

and one more linear inequality enforcing the non-negativity of $F_i$,

$$\sum_{k=1}^K p_k^\Delta r_k(A_{i:}^k x - b_i^k) - r_{K+1}(A(p^c)_{i:} x - b(p^c)_i) \ge 0.$$

Of the $K + 2$ inequalities that define $P_{i,r}$, only the latter one is a boundary of $S_i$. $\qquad\square$

**Parametrization of coefficients** Consider the affine linear map

$$L \colon \mathbb{R}^{\mathcal{O} \times \mathcal{A}} \to \mathbb{R}^{\mathcal{S} \times \mathcal{S}} \times \mathbb{R}^{\mathcal{S}}; \quad p \mapsto (A(p), b(p))$$

defined by (5)–(6) as

$$A(p) = A^0 + \sum_{(o,a) \in \mathcal{O} \times \mathcal{A}} A^{(o,a)} p_{o,a}, \quad \text{where}$$

$$(A^0)_{s,s'} = I_{s,s'},$$

$$(A^{(o,a)})_{s,s'} = -\gamma \alpha(s, a; s') \beta(s; o), \quad (o, a) \in \mathcal{O} \times \mathcal{A},$$

and

$$b(p) = b^0 + \sum_{(o,a) \in \mathcal{O} \times \mathcal{A}} b^{(o,a)} p_{o,a}, \quad \text{where}$$

$$(b^0)_s = 0,$$

$$(b^{(o,a)})_s = r(s, a) \beta(s; o), \quad (o, a) \in \mathcal{O} \times \mathcal{A}.$$

Denote the image of the policy polytope under this map as $S := L(\Delta_{\mathcal{A}}^{\mathcal{O}})$.

We want to represent the same image set $S$ as the intersection of the images of affine linear maps evaluated on hyperrectangles. Suppose that $\mathcal{A} = \{a_1, \ldots, a_{|\mathcal{A}|}\}$ and let $\mathcal{A}_i := \mathcal{A} \setminus \{a_i\}$, $i = 1, \ldots, |\mathcal{A}|$. For each $i$ consider the map

$$L_i \colon [0, 1]^{\mathcal{O} \times \mathcal{A}_i} \to \mathbb{R}^{\mathcal{S} \times \mathcal{S}} \times \mathbb{R}^{\mathcal{S}}; \quad v \mapsto (B_i(v), c_i(v))$$

defined by

$$B_i(v) = B_i^0 + \sum_{(o,a) \in \mathcal{O} \times \mathcal{A}_i} B_i^{(o,a)} v_{o,a}, \quad \text{where}$$

$$(B_i^0)_{s,s'} = (A^0)_{s,s'} + \sum_{o \in \mathcal{O}} (A^{(o,a_i)})_{s,s'},$$

$$(B_i^{(o,a)})_{s,s'} = (A^{(o,a)})_{s,s'} - (A^{(o,a_i)})_{s,s'}, \ (o,a) \in \mathcal{O} \times \mathcal{A}_i,$$

and

$$c_i(v) = c_i^0 + \sum_{(o,a) \in \mathcal{O} \times \mathcal{A}_i} c_i^{(o,a)} v_{o,a}, \quad \text{where}$$

$$(c_i^0)_s = \sum_{o \in \mathcal{O}} (b^{(o,a_i)})_s,$$

$$(c_i^{(o,a)})_s = (b^{(o,a)})_s - (b^{(o,a_i)})_s, \ (o,a) \in \mathcal{O} \times \mathcal{A}_i.$$

Denote the image as $S_i := L_i([0,1]^{\mathcal{O} \times \mathcal{A}_i})$. This can be represented as the image of $L$ composed with an affine map on a hyperrectangle as follows.

**Proposition A.1.** *We have that $S_i = L_i\big([0,1]^{\mathcal{O} \times \mathcal{A}_i}\big) = L(E_i)$, where $E_i := \ell_i([0,1]^{\mathcal{O} \times \mathcal{A}_i}) \subseteq \mathbb{R}^{\mathcal{O} \times \mathcal{A}}$ is the image of the affine map $\ell_i \colon [0,1]^{\mathcal{O} \times \mathcal{A}_i} \to \mathbb{R}^{\mathcal{O} \times \mathcal{A}}; \ v \mapsto p$ defined by*

$$p_{o,a_i} = 1 - \sum_{a \neq a_i} v_{o,a} \quad \text{and} \quad p_{o,a} = v_{o,a}, \text{ for } a \neq a_i.$$

*Proof.* Indeed, if $a \neq a_i$, set $p_{o,a} = v_{o,a}$, and set $p_{o,a_i} = 1 - \sum_{a \neq a_i} v_{o,a}$. Then

$$\sum_{a \in \mathcal{A}} A^{(o,a)} p_{o,a} = A^{(o,a_i)} + \sum_{a \neq a_i} \big(A^{(o,a)} - A^{(o,a_i)}\big) v_{o,a},$$

and similarly

$$\sum_{a \in \mathcal{A}} b^{(o,a)} p_{o,a} = b^{(o,a_i)} + \sum_{a \neq a_i} \big(b^{(o,a)} - b^{(o,a_i)}\big) v_{o,a}.$$

Therefore $S_i = L_i\big([0,1]^{\mathcal{O} \times \mathcal{A}_i}\big) = L(E_i)$. □

Next we show that, under an intersection preservation condition, the set $S$ is the intersection of $S_1, \ldots, S_{|\mathcal{A}|}$.

**Proposition 3.7.** *Assume that $L(E_i) \cap L(E_j) = L(E_i \cap E_j)$ for all $i, j$, or, more strongly, that $L$ is injective on $\bigcup_i E_i$. Then $S = \bigcap_i S_i$.*

*Proof of Proposition 3.7.* Note that

$$\Delta_{\mathcal{A}}^{\mathcal{O}} = \bigcap_i E_i.$$

Indeed, each $E_i$ imposes the constraints $\sum_{a \in \mathcal{A}} p_{o,a} = 1$ and $p_{o,a} \in [0,1]$ for all $a \neq a_i$. Intersecting over all $i$ therefore imposes $p_{o,a} \in [0,1]$ for every $a$, together with the normalization constraint. Hence the intersection is exactly $\Delta_{\mathcal{A}}^{\mathcal{O}}$.

It follows immediately that

$$S = L(\Delta_{\mathcal{A}}^{\mathcal{O}}) = L\left(\bigcap_i E_i\right).$$

Since $\Delta_{\mathcal{A}}^{\mathcal{O}} \subseteq E_i$ for every $i$, we get

$$S \subseteq \bigcap_i L(E_i) = \bigcap_i S_i.$$

For the reverse inclusion, let

$$y \in \bigcap_i S_i.$$

Then $y \in L(E_i)$ for every $i$. By the assumed intersection-preservation property,

$$y \in \bigcap_i L(E_i) = L\left(\bigcap_i E_i\right).$$

Therefore

$$y \in L(\Delta_{\mathcal{A}}^{\mathcal{O}}) = S.$$

Thus

$$\bigcap_i S_i \subseteq S.$$

Combining both inclusions gives $S = \bigcap_i S_i$. □

The above proposition uses an intersection preservation condition. Next we discuss when this condition holds. Let $T \colon \mathbb{R}^{\mathcal{O} \times \mathcal{A}} \to \mathbb{R}^{\mathcal{S} \times \mathcal{S}} \times \mathbb{R}^{\mathcal{S}}$ be the linear map defined by

$$Tz = \sum_{(o,a) \in \mathcal{O} \times \mathcal{A}} \left(A^{(o,a)}, b^{(o,a)}\right) z_{o,a}.$$

Written explicitly in terms of $\alpha, \beta, r$, this is

$$(Tz)_{A,s,s'} = -\gamma \sum_{(o,a)} \alpha(s,a;s') \beta(s;o) z_{o,a},$$

$$(Tz)_{b,s} = \sum_{(o,a)} r(s,a) \beta(s;o) z_{o,a}.$$

The condition $L(E_i) \cap L(E_j) = L(E_i \cap E_j)$ holds if and only if, for every $p \in E_i$ and $q \in E_j$ with

$$T(p - q) = 0,$$

there exists $z \in \ker T$ such that

$$p + z \in \Delta_{\mathcal{A}}^{\mathcal{O}}.$$

In particular, the condition holds whenever the map $T$ is injective.

We show that when $|\mathcal{O}| \leq |\mathcal{S}|$ and $|\mathcal{A}| \leq |\mathcal{S}| + 1$, the map $T$ is generically injective, and thus the assumption of Proposition 3.7 holds.

**Proposition A.2** (Generic injectivity of the coefficient map). *Assume that $|\mathcal{O}| \leq |\mathcal{S}|$ and $|\mathcal{A}| \leq |\mathcal{S}| + 1$. Then, for generic choices of $\beta$, $\alpha$, and $r$, the map $T$ has trivial kernel and hence is injective.*

*Proof.* Write an element $z \in \mathbb{R}^{\mathcal{O} \times \mathcal{A}}$ as a matrix

$$z = (z_{o,a})_{o \in \mathcal{O}, a \in \mathcal{A}}.$$

For each state $s \in \mathcal{S}$ and action $a \in \mathcal{A}$, define

$$u_{s,a} := \sum_{o \in \mathcal{O}} \beta(s;o) z_{o,a}.$$

Thus, for each fixed $s$, the vector

$$u_s = (u_{s,a})_{a \in \mathcal{A}} \in \mathbb{R}^{\mathcal{A}}$$

records the $\beta$-weighted action coordinates of $z$ at state $s$.

Now define, for each $s \in \mathcal{S}$, the vectors

$$w_{s,a} := \left(-\gamma\alpha(s,a;\cdot),\, r(s,a)\right) \in \mathbb{R}^{|\mathcal{S}|+1}.$$

That is,

$$w_{s,a} = \left(-\gamma\alpha(s,a;s_1),\ldots,-\gamma\alpha(s,a;s_{|\mathcal{S}|}),r(s,a)\right).$$

Then the condition $Tz = 0$ is equivalent to

$$\sum_{a \in \mathcal{A}} u_{s,a}w_{s,a} = 0 \qquad \text{for every } s \in \mathcal{S}.$$

Since $|\mathcal{A}| \leq |\mathcal{S}| + 1$, generic choices of $\alpha$ and $r$ make the vectors

$$\{w_{s,a} : a \in \mathcal{A}\}$$

linearly independent for each fixed $s$. Therefore,

$$\sum_{a \in \mathcal{A}} u_{s,a}w_{s,a} = 0$$

implies

$$u_{s,a} = 0 \qquad \text{for all } a \in \mathcal{A}.$$

Since this holds for every $s$, we have

$$\sum_{o \in \mathcal{O}} \beta(s;o)z_{o,a} = 0 \qquad \text{for all } s \in \mathcal{S},\ a \in \mathcal{A}.$$

Now fix an action $a$. The vector

$$z_{\cdot,a} = (z_{o,a})_{o \in \mathcal{O}}$$

satisfies

$$\beta z_{\cdot,a} = 0.$$

Because $|\mathcal{O}| \leq |\mathcal{S}|$, a generic matrix

$$\beta \in \mathbb{R}^{\mathcal{S} \times \mathcal{O}}$$

has full column rank. Hence

$$z_{\cdot,a} = 0.$$

This holds for every action $a$, and therefore

$$z = 0.$$

Thus $\ker T = \{0\}$.

The asserted genericity follows because the failure of full column rank of $\beta$, and the failure of linear independence of each collection $\{w_{s,a} : a \in \mathcal{A}\}$, are algebraic conditions given by the vanishing of suitable minors. Since these minors are not identically zero, their nonvanishing defines a Zariski open dense, hence generic, set of parameters. $\qquad\square$

# B. Reduction to MDPs

Theorem 3.9 describes the solution set in terms of infinitely many piecewise linear inequalities. We show that in the special case of a fully observable MDP, this description consists only of finitely many piecewise linear inequalities.

**Corollary B.1** (Value functions of fully observable MDPs)**.** *Consider a finite discounted MDP with state space $\mathcal{S}$, action space $\mathcal{A} = \{a_1, \ldots, a_{|\mathcal{A}|}\}$, transition kernel $\alpha(s, a; s')$, reward function $r(s, a)$, and discount factor $\gamma \in (0, 1)$.*

*Assume that the hypothesis of Proposition 3.7 holds for the fully observable specialization*

$$\mathcal{O} = \mathcal{S}, \qquad \beta(s; o) = \mathbf{1}_{\{o=s\}}.$$

*For $x \in \mathbb{R}^{\mathcal{S}}$, define*

$$Q_a^x(s) := r(s, a) + \gamma \sum_{s' \in \mathcal{S}} \alpha(s, a; s') x_{s'}.$$

*Then $x \in \mathbb{R}^{\mathcal{S}}$ is a feasible value function, meaning that there exists a policy $\pi \in \Delta_{\mathcal{A}}^{\mathcal{S}}$ such that*

$$x_s = \sum_{a \in \mathcal{A}} \pi_{s,a} \left( r(s, a) + \gamma \sum_{s' \in \mathcal{S}} \alpha(s, a; s') x_{s'} \right) \qquad \text{for all } s \in \mathcal{S},$$

*if and only if*

$$x_s \in \operatorname{conv}\{Q_a^x(s) : a \in \mathcal{A}\} \qquad \text{for every } s \in \mathcal{S}.$$

*Equivalently,*

$$\min_{a \in \mathcal{A}} Q_a^x(s) \leq x_s \leq \max_{a \in \mathcal{A}} Q_a^x(s) \qquad \text{for every } s \in \mathcal{S}.$$

Note that the last expression is a pair of piecewise linear inequalities for each $s$.

*Proof.* We derive the statement as a specialization of Theorem 3.9. In the fully observable case, we take

$$\mathcal{O} = \mathcal{S}, \qquad \beta(s; o) = \mathbf{1}_{\{o=s\}}.$$

Thus the policy parameters $p_{o,a}$ become the usual state-action policy parameters, which we write as $p_{s,a} = \pi_{s,a}$.

**Specialization of the coefficients parametrization**   For each anchor action $a_i$, the matrices in (7)–(8) specialize as follows. First, using

$$(A^0)_{s,s'} = \mathbf{1}_{\{s=s'\}}, \qquad (A^{(o,a)})_{s,s'} = -\gamma \alpha(s, a; s') \mathbf{1}_{\{o=s\}},$$

we obtain

$$\begin{aligned} (B_i^0)_{s,s'} &= (A^0)_{s,s'} + \sum_{o \in \mathcal{S}} (A^{(o,a_i)})_{s,s'} \\ &= \mathbf{1}_{\{s=s'\}} - \gamma \alpha(s, a_i; s'). \end{aligned}$$

Similarly, for $a \neq a_i$,

$$\begin{aligned} (B_i^{(o,a)})_{s,s'} &= (A^{(o,a)})_{s,s'} - (A^{(o,a_i)})_{s,s'} \\ &= -\gamma \big(\alpha(s, a; s') - \alpha(s, a_i; s')\big) \mathbf{1}_{\{o=s\}}. \end{aligned}$$

For the right-hand side, using

$$(b^0)_s = 0, \qquad (b^{(o,a)})_s = r(s, a) \mathbf{1}_{\{o=s\}},$$

we obtain

$$\begin{aligned} (c_i^0)_s &= \sum_{o \in \mathcal{O}} (b^{(o,a_i)})_s \\ &= \sum_{o \in \mathcal{S}} r(s, a_i) \mathbf{1}_{\{o=s\}} = r(s, a_i), \end{aligned}$$

and

$$(c_i^{(o,a)})_s = (b^{(o,a)})_s - (b^{(o,a_i)})_s$$
$$= (r(s,a) - r(s,a_i))\mathbf{1}_{\{o=s\}}.$$

Now fix $x \in \mathbb{R}^{\mathcal{S}}$. Define

$$d_{s,a}^{(i)} := Q_a^x(s) - Q_{a_i}^x(s), \qquad a \neq a_i.$$

From the preceding formulas, the vector

$$B_i^{(o,a)}x - c_i^{(o,a)} = -\mathbf{1}_{\{o=s\}}\left(\sum_{s'}\gamma\big(\alpha(s,a;s') - \alpha(s,a_i;s')\big)x_{s'}\right)_s - \mathbf{1}_{\{o=s\}}\big(r(s,a) - r(s,a_i)\big)_s$$

$$= -\mathbf{1}_{\{o=s\}}\bigg(\underbrace{\big(r(s,a) + \gamma\sum_{s'}\alpha(s,a;s')x_{s'}\big)}_{Q_a^x(s)} - \underbrace{\big(r(s,a_i) + \gamma\sum_{s'}\alpha(s,a_i;s')x_{s'}\big)}_{Q_{a_i}^x(s)}\bigg)_s$$

is supported only on the coordinate $o$, and this coordinate is $-d_{o,a}^{(i)}$.

Moreover,

$$\big(B_i(v^c)x - c_i(v^c)\big)_s = \left(\left(B_i^0 + \sum_{(o,a)\in\mathcal{O}\times\mathcal{A}_i} B_i^{(o,a)}v_{o,a}^c\right)x\right)_s - \left(c_i^0 + \sum_{(o,a)\in\mathcal{O}\times\mathcal{A}_i} c_i^{(o,a)}v_{o,a}^c\right)_s$$

$$= \sum_{s'\in\mathcal{S}}\big(\mathbf{1}_{\{s=s'\}} - \gamma\alpha(s,a_i;s')\big)x_{s'} - r(s,a_i)$$

$$+ \frac{1}{2}\sum_{(o,a)\in\mathcal{S}\times\mathcal{A}_i}\left[\sum_{s'\in\mathcal{S}}\big(-\gamma(\alpha(s,a;s') - \alpha(s,a_i;s'))\mathbf{1}_{\{o=s\}}\big)x_{s'} - \big(r(s,a) - r(s,a_i)\big)\mathbf{1}_{\{o=s\}}\right]$$

$$= x_s - \left(r(s,a_i) + \gamma\sum_{s'\in\mathcal{S}}\alpha(s,a_i;s')x_{s'}\right)$$

$$- \frac{1}{2}\sum_{a\neq a_i}\left[r(s,a) - r(s,a_i) + \gamma\sum_{s'\in\mathcal{S}}\big(\alpha(s,a;s') - \alpha(s,a_i;s')\big)x_{s'}\right]$$

$$= x_s - Q_{a_i}^x(s) - \frac{1}{2}\sum_{a\neq a_i} d_{s,a}^{(i)}.$$

**Specialization of the inequalities in Theorem 3.9** By Theorem 3.9, $x$ is a feasible value function if and only if, for every $y \in \mathbb{R}^{\mathcal{S}}$ and every $i \in \{1, \ldots, |\mathcal{A}|\}$,

$$y^{\top}\big(B_i(v^c)x - c_i(v^c)\big) \leq \frac{1}{2}\sum_{(o,a)\in\mathcal{S}\times\mathcal{A}_i}|y^{\top}\big(B_i^{(o,a)}x - c_i^{(o,a)}\big)|.$$

Using the specialization above, this becomes

$$\sum_{s\in\mathcal{S}} y_s\left(x_s - Q_{a_i}^x(s) - \frac{1}{2}\sum_{a\neq a_i} d_{s,a}^{(i)}\right) \leq \frac{1}{2}\sum_{s\in\mathcal{S}}|y_s|\sum_{a\neq a_i}|d_{s,a}^{(i)}|.$$

Since this inequality must hold for all $y$, and also for $-y$, it is equivalent to the collection of scalar inequalities

$$\left|x_s - Q_{a_i}^x(s) - \frac{1}{2}\sum_{a\neq a_i}\big(Q_a^x(s) - Q_{a_i}^x(s)\big)\right| \leq \frac{1}{2}\sum_{a\neq a_i}|Q_a^x(s) - Q_{a_i}^x(s)|$$

for every $s \in \mathcal{S}$ and every $i$.

**Simplification of the inequalities**    It remains to simplify these scalar inequalities. Fix a state $s$ and write $q_a = Q_a^x(s)$ and $z = x_s$. For a fixed anchor action $a_i$, the inequality is

$$|z - q_{a_i} - \frac{1}{2} \sum_{a \neq a_i} (q_a - q_{a_i})| \leq \frac{1}{2} \sum_{a \neq a_i} |q_a - q_{a_i}|.$$

By Proposition B.2, this is equivalent to

$$z \in \left[ q_{a_i} + \sum_{a:q_a < q_{a_i}} (q_a - q_{a_i}), \; q_{a_i} + \sum_{a:q_a > q_{a_i}} (q_a - q_{a_i}) \right].$$

**Intersection of the parametric systems**    Intersecting over all anchor actions $a_i$, this interval condition is equivalent to

$$\min_{a \in \mathcal{A}} q_a \leq z \leq \max_{a \in \mathcal{A}} q_a.$$

Indeed, the anchor action attaining the minimum forces $z \geq \min_a q_a$, and the anchor action attaining the maximum forces $z \leq \max_a q_a$. Conversely, if $z \in [\min_a q_a, \max_a q_a]$, then the above interval condition holds for every anchor action.

Returning to the original notation, we have shown that the inequalities in Theorem 3.9 are equivalent to

$$\min_{a \in \mathcal{A}} Q_a^x(s) \leq x_s \leq \max_{a \in \mathcal{A}} Q_a^x(s) \qquad \text{for every } s \in \mathcal{S}.$$

Since the convex hull of finitely many real numbers is precisely the interval between their minimum and maximum, this is equivalent to

$$x_s \in \mathrm{conv}\{Q_a^x(s) : a \in \mathcal{A}\} \qquad \text{for every } s \in \mathcal{S}.$$

Finally, the condition $x_s \in \mathrm{conv}\{Q_a^x(s) : a \in \mathcal{A}\}$ is equivalent to the existence, for each state $s$, of coefficients $\pi_{s,a} \geq 0$ with $\sum_a \pi_{s,a} = 1$ such that

$$x_s = \sum_{a \in \mathcal{A}} \pi_{s,a} Q_a^x(s) = \sum_{a \in \mathcal{A}} \pi_{s,a} \left( r(s,a) + \gamma \sum_{s' \in \mathcal{S}} \alpha(s,a;s') x_{s'} \right).$$

This is precisely the Bellman equation for the policy $\pi \in \Delta_{\mathcal{A}}^{\mathcal{S}}$. Hence the desired characterization follows.    $\square$

**Proposition B.2.** *The inequality*

$$|z - q_{a_i} - \frac{1}{2} \sum_{a \neq a_i} (q_a - q_{a_i})| \leq \frac{1}{2} \sum_{a \neq a_i} |q_a - q_{a_i}|$$

*is equivalent to*

$$z \in \left[ q_{a_i} + \sum_{a:q_a < q_{a_i}} (q_a - q_{a_i}), \; q_{a_i} + \sum_{a:q_a > q_{a_i}} (q_a - q_{a_i}) \right].$$

*Proof.* Indeed, set

$$\delta_a := q_a - q_{a_i}, \qquad a \neq a_i.$$

Then the inequality becomes

$$|z - q_{a_i} - \frac{1}{2} \sum_{a \neq a_i} \delta_a| \leq \frac{1}{2} \sum_{a \neq a_i} |\delta_a|.$$

This is equivalent to

$$-\frac{1}{2} \sum_{a \neq a_i} |\delta_a| \leq z - q_{a_i} - \frac{1}{2} \sum_{a \neq a_i} \delta_a \leq \frac{1}{2} \sum_{a \neq a_i} |\delta_a|.$$

Equivalently,

$$q_{a_i} + \frac{1}{2} \sum_{a \neq a_i} \delta_a - \frac{1}{2} \sum_{a \neq a_i} |\delta_a| \leq z \leq q_{a_i} + \frac{1}{2} \sum_{a \neq a_i} \delta_a + \frac{1}{2} \sum_{a \neq a_i} |\delta_a|.$$

Now use

$$\frac{1}{2}(\delta_a - |\delta_a|) = \begin{cases} \delta_a, & \delta_a < 0, \\ 0, & \delta_a \geq 0, \end{cases} \qquad \frac{1}{2}(\delta_a + |\delta_a|) = \begin{cases} 0, & \delta_a \leq 0, \\ \delta_a, & \delta_a > 0. \end{cases}$$

The lower endpoint becomes

$$q_{a_i} + \sum_{a \neq a_i} \frac{1}{2}(\delta_a - |\delta_a|) = q_{a_i} + \sum_{a:q_a < q_{a_i}} (q_a - q_{a_i}),$$

and the upper endpoint becomes

$$q_{a_i} + \sum_{a \neq a_i} \frac{1}{2}(\delta_a + |\delta_a|) = q_{a_i} + \sum_{a:q_a > q_{a_i}} (q_a - q_{a_i}).$$

Hence the absolute-value inequality is equivalent to

$$z \in \left[ q_{a_i} + \sum_{a:q_a < q_{a_i}} (q_a - q_{a_i}),\ q_{a_i} + \sum_{a:q_a > q_{a_i}} (q_a - q_{a_i}) \right].$$

$\square$

## C. Proofs of Results in Section 4

We first show the equivalent formulation.

**Proposition 4.1.** *A vector $x$ solves a parametric matrix system $A(p)x - b(p) = 0$ with parametrization (4) if and only if there exist $q_k \in [-1, 1]$, $k = 1, \ldots, K$ such that*

$$A(p^c)x - b(p^c) = \sum_{k=1}^{K} q_k p_k^\Delta (A^k x - b^k).$$

*Proof.* We start by deriving a necessary and sufficient condition for $x$ to solve a parametric matrix equation $A(p)x - b(p) = 0$ in terms of the quantities $A(p^c), b(p^c), p_k^\Delta, A^k, b^k$. The parametric matrices under consideration depend affinely on the parameters $p_k$ via

$$A(p) = A^0 + \sum_{k=1}^{K} p_k A^k, \quad b(p) = b^0 + \sum_{k=1}^{K} p_k b^k.$$

At the parameter midpoint $p^c$ they are given by $A(p^c) = A^0 + \sum_{k=1}^{K} p_k^c A^k$, $b(p) = b^0 + \sum_{k=1}^{K} p_k^c b^k$. A general $p$ in the interval vector $[p]$ can be expressed as $p = p^c + \delta p$, i.e., the midpoint plus some deviation. Thus we can write the parametric coefficients at any arbitrary $p$ as

$$A(p) = A(p^c + \delta p) = A^0 + \sum_{k=1}^{K} (p_k^c + \delta p_k) A^k$$

$$= A(p^c) + \sum_{k=1}^{K} \delta p_k A^k,$$

$$b(p) = b(p^c + \delta p) = b^0 + \sum_{k=1}^{K} (p_k^c + \delta p_k) b^k$$

$$= b(p^c) + \sum_{k=1}^{K} \delta p_k b^k.$$

Let $q_k \in [-1, 1]$ be the factor for each parameter component, such that $\delta p_k = q_k p^\Delta$. Then

$$A(p) = A(p^c) + \sum_{k=1}^{K} q_k p_k^\Delta A^k$$

$$b(p) = b(p^c) + \sum_{k=1}^{K} q_k p_k^\Delta b^k.$$

Therefore we can rewrite $A(p)x - b(p) = 0$ as

$$A(p)x - b(p) = A(p^c)x + \sum_{k=1}^{K} q_k p_k^\Delta A^k x - (b(p^c) + \sum_{k=1}^{K} q_k p_k^\Delta b^k),$$

$$= A(p^c)x - b(p^c) + \sum_{k=1}^{K} q_k p_k^\Delta (A^k x - b^k).$$

Thus $x$ solves $A(p)x - b(p) = 0$ for some $p \in [p]$ iff there exist $q_k \in [-1, 1]$ such that

$$A(p^c)x - b(p^c) = \sum_{k=1}^{K} q_k p_k^\Delta (A^k x - b^k).$$

$\square$

Recall that a polyhedron $P \subseteq \mathbb{R}^n$ has a unique and minimal description in terms of a finite number of linear inequalities, i.e., $P \subseteq \mathbb{R}^n = \{x \in \mathbb{R}^n : a_i x \leq b_i, i = 1, \ldots, m\}$. Each of the inequalities in the unique and minimal description of $P$ defines a *facet* of $P$, a face of the polyhedron which has dimension one less than the dimension of $P$ (Wolsey, 2020, Proposition 9.2). For centrally symmetric polyhedra, including zonotopes of the form $Z(g_1, \ldots, g_K) = \sum_{k=1}^{K} [-g_k, g_k]$, facet defining inequalities can be characterized as follows.

**Proposition C.1** (Facet inequalities for a zonotope). *Let $g_1, \ldots, g_K \in \mathbb{R}^d$ and define the centrally symmetric zonotope*

$$Z(g_1, \ldots, g_K) = \sum_{k=1}^{K} [-g_k, g_k] = \left\{ \sum_{k=1}^{K} \lambda_k g_k \; : \; -1 \leq \lambda_k \leq 1 \right\}.$$

*Then $h \in Z(g_1, \ldots, g_K)$ if and only if*

$$w^\top h \leq \sum_{k=1}^{K} |w^\top g_k|$$

*for every facet normal $w$ of the zonotope.*

*Equivalently, it is sufficient to verify the inequality for all nonzero vectors $w$ orthogonal to a linearly independent collection of generators $\{g_i : i \in I\}$ whose span has codimension one and determines a facet direction of the zonotope, that is,*

$$w^\top g_i = 0 \qquad \text{for all } i \in I.$$

*Proof.* The support function of the zonotope is

$$\sigma_Z(w) = \max_{z \in Z} w^\top z.$$

Since $Z = \sum_{k=1}^{K} [-g_k, g_k]$, support functions add under Minkowski sums. Moreover,

$$\max_{\lambda_k \in [-1,1]} w^\top (\lambda_k g_k) = |w^\top g_k|.$$

Therefore

$$\sigma_Z(w) = \sum_{k=1}^{K} |w^\top g_k|.$$

For any compact convex set $K \subseteq \mathbb{R}^d$, one has the standard halfspace representation

$$K = \{h \in \mathbb{R}^d : w^\top h \leq \sigma_K(w) \text{ for all } w \in \mathbb{R}^d\}.$$

Applying this to $Z$ gives

$$h \in Z \quad \Longleftrightarrow \quad w^\top h \leq \sum_{k=1}^{K} |w^\top g_k| \text{ for all } w \in \mathbb{R}^d.$$

Since $Z$ is a polytope, it is enough to check only the irredundant supporting halfspaces, namely the facet-defining inequalities. The normal fan of a zonotope is the central hyperplane arrangement with hyperplanes $g_k^\perp$. Hence a facet normal $w$ lies on a one-dimensional cone of this arrangement. Equivalently, $w$ is orthogonal to a maximal collection of linearly independent generators spanning the corresponding facet direction. For such a normal, the associated facet inequality is precisely

$$w^\top h \leq \sigma_Z(w) = \sum_{k=1}^{K} |w^\top g_k|.$$

Thus membership in $Z$ is equivalent to satisfying all such facet-defining inequalities. $\qquad\square$

**Corollary 4.3.** *For any POMDP, the solution set to the Bellman equation, which is the set of feasible value functions, is a semi-algebraic set.*

*Proof of Corollary 4.3.* From Theorem 4.2, we can obtain a finite description of the solution set to the Bellman equation for any POMDP.

As long as our finite description consists of finitely many polynomial inequalities and equalities, then we have a semi-algebraic set. We note moreover that the finite description obtained from Theorem 4.2 requires the existence of the vectors $q^{(i)}$. In this sense, the solution set of the Bellman equation, together with the $q^{(i)}$-vectors, is a semi-algebraic set.

However, by the Tarski-Seidenberg Theorem, we may eliminate existential quantifiers in the construction of a semi-algebraic set and still retain a semi-algebraic set (Drton & Sullivant, 2007). Put another way, the image of the semi-algebraic set $A(x,q) = \{x, q : D_1 q^{(1)} = C_1, -1 \leq q^{(1)} \leq 1, D_2 q^{(2)} = C_2, -1 \leq q^{(2)} \leq 1\}$ under the projection map $A(x,q) \rightarrow \tilde{A}(x)$ is itself semi-algebraic.

It remains to be shown that the description provided in Theorem 4.2 indeed consists of finitely many polynomial inequalities and equalities. By Theorem 4.2, the solution set to the Bellman equation admits a finite description using auxiliary vectors $q^{(i)}$ of the form

$$D_i(x)\, q^{(i)} = C_i(x), -1 \leq q_j^{(i)} \leq 1,$$

for $i$ and all components $j$. There are finitely many such equalities and inequalities.

Each entry of $D_i(x)$ and $C_i(x)$ is a polynomial function of $x$, since they are constructed from the POMDP using finitely many sums and products. Hence each constraint $D_i(x)q^{(i)} = C_i(x)$ is a system of polynomial equalities in the joint variables $(x, q)$, and the box constraints on $q^{(i)}$ are polynomial inequalities. Therefore the set

$$A = \left\{ (x,q) : D_i(x)q^{(i)} = C_i(x), \ -1 \leq q^{(i)} \leq 1 \right\}$$

is semi-algebraic.

The feasible value-function set is precisely the projection of $A$ onto the $x$-coordinates:

$$\tilde{A} = \{x : \exists q \text{ such that } (x, q) \in A\}.$$

By the Tarski–Seidenberg theorem, projections of semi-algebraic sets are semi-algebraic. Hence $\tilde{A}$ is semi-algebraic. $\square$

## D. Semi-algebraic Description of the Solution Set for the Bellman Equation

**Theorem 4.4** (Semi-algebraic description of the solution set to the Bellman equation). *Consider a POMDP with finite state space $\mathcal{S}$, observation space $\mathcal{O}$, and action space $\mathcal{A}$. Fix orderings of $\mathcal{O}$ and $\mathcal{A}$, and write a stochastic memoryless policy as a vector $P \in \mathbb{R}^{|\mathcal{O}||\mathcal{A}|}$ with entries satisfying $P(a|o) \geq 0$, $\sum_{a \in \mathcal{A}} P(a|o) = 1$ for all $o \in \mathcal{O}$.*

*For each $(o, a) \in \mathcal{O} \times \mathcal{A}$, define $u_{o,a}(x) \in \mathbb{R}^{|\mathcal{S}|}$ by*

$$u_{o,a}(x)_s = \beta(s; o) \left( \gamma \sum_{s' \in \mathcal{S}} \alpha(s, a; s') x_{s'} + r(s, a) \right).$$

*Let $d(x) = x \in \mathbb{R}^{|\mathcal{S}|}$, and let $R \in \mathbb{R}^{|\mathcal{O}| \times |\mathcal{O}||\mathcal{A}|}$ be the matrix encoding the row-sum constraints, so that $(RP)_o = \sum_{a \in \mathcal{A}} P(a|o)$.*

*Define $M(x) \in \mathbb{R}^{|\mathcal{S}| \times |\mathcal{O}||\mathcal{A}|}$ as the matrix whose columns are $u_{o,a}(x)$, and set*

$$C(x) = \begin{pmatrix} M(x) \\ R \end{pmatrix}, \qquad f(x) = \begin{pmatrix} d(x) \\ \mathbf{1}_{|\mathcal{O}|} \end{pmatrix}.$$

*Then $x$ is a feasible value function if and only if there exists $P \in \mathbb{R}^{|\mathcal{O}||\mathcal{A}|}$ such that $C(x)P = f(x), P \geq 0$.*

*Moreover, the solution set $S$ admits the following quantifier-free semi-algebraic description as a finite union stratified by* rank $C(x) = \rho$:

$$S = \bigcup_{\rho=1}^{|\mathcal{S}|+|\mathcal{O}|} \bigcup_{\substack{I \subseteq \{1,\ldots,|\mathcal{S}|+|\mathcal{O}|\} \\ |I|=\rho}} \bigcup_{\substack{B \subseteq \{1,\ldots,|\mathcal{O}||\mathcal{A}|\} \\ |B|=\rho}} S_{\rho,I,B},$$

*where $S_{\rho,I,B}$ is the set of all $x \in \mathbb{R}^{\mathcal{S}}$ satisfying the following polynomial inequalities:*

 (i)  $\det C_{I,B}(x) \neq 0$ *and every* $(\rho + 1) \times (\rho + 1)$ *minor of* $C(x)$ *vanishes;*

 (ii) *every* $(\rho + 1) \times (\rho + 1)$ *minor of the augmented matrix* $[C(x)|f(x)]$ *vanishes;*

(iii) *for all* $t = 1, \ldots, \rho$, $\det C_{I,B,t}(x) \det C_{I,B}(x) \geq 0$.

*Here, for index sets $I \subseteq \{1, \ldots, |\mathcal{S}| + |\mathcal{O}|\}$ and $B \subseteq \{1, \ldots, |\mathcal{O}||\mathcal{A}|\}$ with $|I| = |B| = \rho$, the matrix $C_{I,B}(x)$ denotes the $\rho \times \rho$ submatrix of $C(x)$ with row set $I$ and column set $B$. $C_{I,B,t}(x)$ is obtained from $C_{I,B}(x)$ by replacing its $t$-th column by $f_I(x)$, the corresponding subvector of $f(x)$.*

*Proof of Theorem 4.4.* The proof proceeds in four steps:

- In Step 1, we rewrite the Bellman equation as a parametric linear feasibility system in the policy parameters.

- In Step 2, we show that feasibility of this system is equivalent to feasibility of a full-row-rank subsystem obtained by selecting a maximal independent row set; the algebraic conditions that identify such a subsystem and certify consistency are exactly (i)–(ii).

- In Step 3, we apply support reduction to the reduced subsystem to obtain a basic feasible solution.

- In Step 4, we eliminate the policy parameters via Cramer's rule, yielding the sign-polynomial conditions (iii), and combine both directions to obtain the stratified union.

- In Step 5, we show via forward and backward inclusions that rank stratification indeed yields the solution set $S = \cup_\rho \cup_I \cup_B S_{\rho,I,B}$.

Throughout, we vectorize $P$ by $p_{(o-1)|\mathcal{A}|+a} = P(a|o)$, so $p \in \mathbb{R}^{|\mathcal{O}||\mathcal{A}|}$.

**Step 1: Bellman equation as a parametric linear system.** Recall the effective policy, or policy over observations, obtained as

$$\pi_s(a) := \sum_{o \in \mathcal{O}} \beta(s; o)\, P(a|o),$$

the policy-weighted transition kernel $P^\pi_{s,s'} = \sum_a \pi_s(a)\, \alpha(s, a; s')$, and the expected reward $r^\pi_s = \sum_a \pi_s(a)\, r(s, a)$. The Bellman optimality equation is $(I - \gamma P^\pi)x = r^\pi$. Rearranging and expanding componentwise, we get

$$(I - \gamma P^\pi)r^\pi = 0$$

$$x_s - \gamma \sum_a \pi_s(a) \sum_{s'} \alpha(s, a; s')x_{s'} - \sum_a \pi_s(a)\, r(s, a) = 0$$

$$x_s - \sum_{o,a} \beta(s; o)\, P(a|o)\left(\gamma \sum_{s'} \alpha(s, a; s')x_{s'} + r(s, a)\right) = 0$$

$$x_s - \sum_{o,a} u_{o,a}(x)_s\, P(a|o) = 0$$

$$\sum_{o,a} u_{o,a}(x)_s\, P(a|o) = x_s,$$

with $u_{o,a}(x)$ as in the theorem. For fixed $x$, this is linear in $P$. Letting $M(x)$ be the matrix with columns $u_{o,a}(x)$ and $d(x) = x$, the Bellman equation is equivalent to $M(x)p = d(x)$.

The row-sum constraints $\sum_a P(a|o) = 1$ for each $o \in \mathcal{O}$ are encoded by $Rp = \mathbf{1}_{|\mathcal{O}|}$, where $R \in \mathbb{R}^{|\mathcal{O}| \times |\mathcal{O}||\mathcal{A}|}$ has $R_{o,(o-1)|\mathcal{A}|+a} = 1$ for $a = 1, \ldots, |\mathcal{A}|$ and zero elsewhere.

Together with $p \geq 0$, the Bellman equation under a stochastic memoryless policy is equivalent to the parametric linear feasibility system

$$C(x)\, p = f(x), \qquad p \geq 0, \tag{12}$$

with $C(x) \in \mathbb{R}^{(|\mathcal{S}|+|\mathcal{O}|) \times |\mathcal{O}||\mathcal{A}|}$ and $f(x) \in \mathbb{R}^{|\mathcal{S}|+|\mathcal{O}|}$ as defined in the theorem.

Hence $x$ is a feasible value function if and only if (12) admits a solution.

**Step 2: Reducing to a full-row-rank subsystem** Note that if $\operatorname{rank} C(x) = |\mathcal{S}| + |\mathcal{O}|$, then $C(x)$ would have full row rank, the equation $C(x)p = f(x)$ is automatically consistent, and we could take a square invertible submatrix and proceed. However, $C(x)$ is in general not square (it has $|\mathcal{S}| + |\mathcal{O}|$ rows and $|\mathcal{O}||\mathcal{A}|$ columns) and need not have full row rank.

Therefore, our goal in this step is to reduce (12) to an equivalent system in which the coefficient matrix has full row rank. We let the rank be a variable $\rho \in \{1, \ldots, |\mathcal{S}| + |\mathcal{O}|\}$ over which we stratify, and for each value of $\rho$ extract the conditions necessary to obtain a full-row-rank subsystem.

Fix $x$ and write $\rho = \operatorname{rank} C(x)$. By definition of rank, there exist row and column index sets $I \subseteq \{1, \ldots, |\mathcal{S}| + |\mathcal{O}|\}$ and $B \subseteq \{1, \ldots, |\mathcal{O}||\mathcal{A}|\}$ with $|I| = |B| = \rho$ such that $\det C_{I,B}(x) \neq 0$, while every $(\rho+1) \times (\rho+1)$ minor of $C(x)$ vanishes. These two conditions together algebraically certify $\operatorname{rank} C(x) = \rho$ and are exactly condition (i).

Equation (12) has a solution if and only if $\operatorname{rank}[C(x)|f(x)] = \operatorname{rank} C(x) = \rho$, i.e., every $(\rho + 1) \times (\rho + 1)$ minor of $[C(x)|f(x)]$ vanishes. This is condition (ii).

Now suppose (i)–(ii) hold for some $(\rho, I, B)$. Since $|I| = \rho = \operatorname{rank} C(x)$ and the rows indexed by $I$ are linearly independent, they form a basis of the row space of $C(x)$. Hence for every row index $j \in \{1, \ldots, |\mathcal{S}| + |\mathcal{O}|\}$, there exists a unique $\lambda_j \in \mathbb{R}^\rho$ such that

$$C_{j,*}(x) = \lambda_j^\top C_{I,*}(x).$$

Since $C_{I,*}(x)$ has full row rank, restricting both sides to columns $B$ gives $C_{j,B}(x) = \lambda_j^\top C_{I,B}(x)$. $C_{I,B}(x)$ is invertible, so we can uniquely obtain $\lambda_j = (C_{I,B}(x)^{-1})^C_{j,B}(x)^\top$

Since the rows of $[C(x)|f(x)]$ indexed by $I$ already contribute $\rho$ independent rows, every other row $[C_{j,*}(x)|f_j(x)]$ must be a linear combination of these, say with coefficients $\mu_j \in \mathbb{R}^\rho$. Restricting to the $C$-block gives $C_{j,*}(x) = \mu_j^\top C_{I,*}(x)$, and by the uniqueness just established, $\mu_j = \lambda_j$. Restricting to the $f$-block then yields

$$f_j(x) = \lambda_j^\top f_I(x) \qquad \text{for every } j \in \{1, \ldots, |\mathcal{S}| + |\mathcal{O}|\}.$$

Consequently, for any $p$ satisfying $C_{I,*}(x)\,p = f_I(x)$, we have $C_{j,*}(x)\,p = \lambda_j^\top C_{I,*}(x)\,p = \lambda_j^\top f_I(x) = f_j(x)$ for every $j$, i.e., $C(x)p = f(x)$. The converse is immediate. Therefore

$$C(x)p = f(x) \quad \Longleftrightarrow \quad C_{I,*}(x)\,p = f_I(x),$$

so feasibility of (12) reduces, under (i)–(ii), to feasibility of the full-row-rank subsystem

$$C_{I,*}(x)\,p = f_I(x), \qquad p \geq 0. \tag{13}$$

**Step 3: Support reduction on the reduced subsystem**  At this point we might hope to be done as we could take any $(I, B)$ certifying $\operatorname{rank} C(x) = \rho$ and set $p_B = C_{I,B}(x)^{-1} f_I(x)$, $p_{B^c} = 0$. By construction this $p$ solves $C_{I,*}(x)p = f_I(x)$, hence by Step 2 it solves (12) as an equality. The problem is the nonnegativity constraint: the $p_B$ we obtain may have entries which are negative.

The purpose of Step 3 is to choose $B$ correctly. We take a known nonnegative solution $p^0$ of (13) (which exists by assumption when $x \in S$) and walk it down to a basic feasible solution $p^\star$ whose support is small enough to be contained in some column set $B$ of size $\rho$ with $C_{I,B}(x)$ invertible. Elements of the solution $C_{I,B}(x)^{-1} f_I(x)$ then equal the support entries of $p^\star$, which are nonnegative by construction.

Concretely, we show that if (13) is feasible, then it admits a basic feasible solution $p^\star$ whose support is contained in a column index set $B$ of size $\rho$ for which $C_{I,B}(x)$ is invertible.

Let $p^0$ be any feasible point of (13) and put $J = \operatorname{supp}(p^0)$. If the columns $\{C_{I,j}(x) : j \in J\}$ are linearly dependent, there exists a nonzero $h \in \mathbb{R}^{|J|}$ with $C_{I,J}(x)\,h = 0$. Extend $h$ to $\bar{h} \in \mathbb{R}^{|\mathcal{O}||\mathcal{A}|}$ by zero outside $J$; then $C_{I,*}(x)\bar{h} = 0$ and $\operatorname{supp}(\bar{h}) \subseteq J$.

Consider $p(t) = p^0 + t\bar{h}$. For every $t$, $C_{I,*}(x)\,p(t) = f_I(x)$, so feasibility reduces to $p(t) \geq 0$. Since $\bar{h}$ is supported on $J$ where $p^0 > 0$, both signs may occur in $\{\bar{h}_j : j \in J\}$, or all nonzero entries may share one sign. In either case, define the feasible interval $[L, U]$ with

$$L = \max_{j:\,\bar{h}_j > 0}\left(-\frac{p_j^0}{\bar{h}_j}\right), \qquad U = \min_{j:\,\bar{h}_j < 0}\left(-\frac{p_j^0}{\bar{h}_j}\right),$$

under the convention $L = -\infty$ if no $\bar{h}_j > 0$ and $U = +\infty$ if no $\bar{h}_j < 0$. Since $p(0) = p^0 \geq 0$ we have $L \leq 0 \leq U$. Because $\bar{h} \neq 0$, at least one of $L, U$ is finite: if any $\bar{h}_j > 0$ then $L > -\infty$, and if any $\bar{h}_j < 0$ then $U < +\infty$, and at least one such $j$ exists. Choose $t^\star$ to be a finite endpoint of $[L, U]$, taking $t^\star = U$ if $U < +\infty$, otherwise $t^\star = L$. At $t = t^\star$, some index $j^\star \in J$ has $p_{j^\star}(t^\star) = 0$: set $p^1 = p(t^\star)$. Then $p^1 \geq 0$, $C_{I,*}(x)p^1 = f_I(x)$, and $\operatorname{supp}(p^1) \subsetneq J$.

If we iterate this reduction, it terminates as the support strictly decreases at each step and will yield a feasible $p^\star$ whose support $J^\star := \operatorname{supp}(p^\star)$ satisfies that the columns $\{C_{I,j}(x) : j \in J^\star\}$ are linearly independent. In particular, $|J^\star| \leq \rho$.

It remains to extend $J^\star$ to a column set $B$ of size exactly $\rho$ with $\det C_{I,B}(x) \neq 0$. Since $\operatorname{rank} C_{I,*}(x) = \rho$, the columns of $C_{I,*}(x)$ span an $\rho$-dimensional space, and the linearly independent set $\{C_{I,j}(x) : j \in J^\star\}$ can be extended to a basis $\{C_{I,j}(x) : j \in B\}$ of this column space, with $J^\star \subseteq B \subseteq \{1, \ldots, |\mathcal{O}||\mathcal{A}|\}$ and $|B| = \rho$. By construction, $C_{I,B}(x)$ has $\rho$ linearly independent columns and $\rho$ rows, so it is invertible: $\det C_{I,B}(x) \neq 0$. This $B$ is precisely a column set appearing in condition (i).

Setting $p_{B^c}^\star = 0$ (which holds since $\operatorname{supp}(p^\star) \subseteq J^\star \subseteq B$), the equation $C_{I,*}(x)p^\star = f_I(x)$ becomes $C_{I,B}(x)\,p_B^\star = f_I(x)$, giving

$$p_B^\star = C_{I,B}(x)^{-1} f_I(x), \qquad p_{B^c}^\star = 0, \qquad p_B^\star \geq 0.$$

Conversely, any $p$ of this form with $p_B \geq 0$ is feasible for (13), hence (by Step 2) for (12).

**Step 4: Construction of the sign polynomial via Cramer's rule**  With $\det C_{I,B}(x) \neq 0$ from (i), Cramer's rule gives

$$(p_B^\star)_t = \frac{\det C_{I,B,t}(x)}{\det C_{I,B}(x)}, \qquad t = 1, \ldots, \rho,$$

where $C_{I,B,t}(x)$ is obtained from $C_{I,B}(x)$ by replacing its $t$-th column with $f_I(x)$.

Because $\det C_{I,B}(x) \neq 0$ by (i), the quantity $(\det C_{I,B}(x))^2$ is strictly positive, so multiplying the inequality $(p_B^\star)_t \geq 0$ through by $(\det C_{I,B}(x))^2$ is sign-preserving. The nonnegativity condition $p_B^\star \geq 0$ is therefore equivalent to the polynomial inequalities

$$\det C_{I,B,t}(x) \det C_{I,B}(x) \geq 0, \qquad t = 1, \ldots, \rho,$$

which is condition (iii).

**Step 5: Forward and Backward Inclusion**  Recall from Step 1 that $S = \{x \in \mathbb{R}^{\mathcal{S}} : \exists p \geq 0, \ C(x)p = f(x)\}$, the set of $x$ for which (12) is feasible.

The set $S_{\rho,I,B}$ is the locus of $x \in \mathbb{R}^{|\mathcal{S}|}$ at which the polynomial conditions (i)–(iii) hold for the given triple $(\rho, I, B)$; each $S_{\rho,I,B}$ is therefore a basic semi-algebraic set, cut out by polynomial equalities (vanishing minors), one polynomial nonvanishing condition ($\det C_{I,B}(x) \neq 0$), and the sign-polynomial inequalities of (iii).

We now combine the four steps to establish that feasibility of (12) is equivalent to membership in this union:

$$S = \bigcup_{\rho=1}^{|\mathcal{S}|+|\mathcal{O}|} \ \bigcup_{|I|=\rho} \ \bigcup_{|B|=\rho} S_{\rho,I,B}.$$

($\Rightarrow$)  To show sufficiency, suppose $x \in S_{\rho,I,B}$ for some triple $(\rho, I, B)$, i.e., conditions (i)–(iii) hold.

By (i), $\det C_{I,B}(x) \neq 0$, so define $p \in \mathbb{R}^{|\mathcal{O}||\mathcal{A}|}$ by $p_B = C_{I,B}(x)^{-1} f_I(x)$ and $p_{B^c} = 0$.

By Cramer's rule and (iii), $p \geq 0$. By construction, $C_{I,*}(x)\, p = f_I(x)$, i.e., $p$ solves the reduced system (13).

By Step 2 (using (i)–(ii)), $p$ also solves (12), so $x \in S$.

($\Leftarrow$)  To show necessity, suppose $x \in S$, so (12) admits some solution $p^0$.

Set $\rho = \operatorname{rank} C(x) \in \{1, \ldots, |\mathcal{S}| + |\mathcal{O}|\}$. By definition of rank, choose any $I, B$ with $|I| = |B| = \rho$ such that $\det C_{I,B}(x) \neq 0$ and every $(\rho+1) \times (\rho+1)$ minor of $C(x)$ vanishes; this gives (i).

Since (12) is feasible, $f(x)$ lies in the column space of $C(x)$, hence $\operatorname{rank}[C(x)|f(x)] = \operatorname{rank} C(x) = \rho$, and every $(\rho+1) \times (\rho+1)$ minor of $[C(x)|f(x)]$ vanishes; this gives (ii).

By Step 2, $p^0$ also solves the reduced system (13). Applying Step 3 to $p^0$ produces a basic feasible solution $p^\star$ supported in some $B' \subseteq \{1, \ldots, |\mathcal{O}||\mathcal{A}|\}$ with $|B'| = \rho$ and $\det C_{I,B'}(x) \neq 0$. Replacing $B$ by $B'$ (which still satisfies (i) since $\det C_{I,B'}(x) \neq 0$ and the vanishing of $(\rho+1) \times (\rho+1)$ minors is a property of $C(x)$ independent of the choice of $B$), Cramer's rule applied to $p^\star$ together with $p^\star \geq 0$ yields (iii).

Hence $x \in S_{\rho,I,B'}$.

Both inclusions give the claimed quantifier-free semi-algebraic description

$$S = \bigcup_{\rho=1}^{|\mathcal{S}|+|\mathcal{O}|} \ \bigcup_{|I|=\rho} \ \bigcup_{|B|=\rho} S_{\rho,I,B}. \qquad \square$$

*Remark* D.1 (Complexity of quantifier elimination in the parametric system). In the main body of the paper, we discussed how the Tarski-Seidenberg theorem guarantees that the solution set to the Bellman equation is a semi-algebraic set. This statement relies on quantifier elimination in that we need to pass from conditions on the solution set which depend on simultaneous choices of $(x, q)$ to one depending only on $x$. This can be done by projection of the solution set onto the $x$ coordinates.

While this use of quantifier elimination is useful for understanding the properties of the solution set to the Bellman equation, it is not a priori a useful observation, as the computational complexity of quantifier elimination is known to be doubly exponential in the number of variables (Basu et al., 2006, Chapter 14).

The result in Theorem 4.4 shows that a quantifier-free parametrization can be achieved with $\binom{k}{l}$ determinantal conditions for

each choice of rank in the stratification, where $k = |\mathcal{O}||\mathcal{A}|$ and $l$ is the rank of the matrix

$$C(x) = \begin{pmatrix} M(x) \\ R \end{pmatrix}.$$

This is substantially lower than the doubly exponential bound cited above and moreover results in tractably computable conditions for a value function to be feasible for the Bellman equation for a given POMDP.

# E. Numerical Illustrations

Code for generating the figures described below can be obtained at the below link:

`https://github.com/ryan-a-anderson/pomdp-value-geometry.`

## E.1. Figure 2, Figure 3

Let $\mathcal{S} = \{0, 1\}$, $\mathcal{A} = \{0, 1\}$, and $\mathcal{O} = \{0, 1, 2\}$. For each action $a \in \mathcal{A}$, the transition matrix $P^a \in \mathbb{R}^{2 \times 2}$ has entries $P^a_{ss'} = \mathbb{P}(s_{t+1} = s' | s_t = s, a_t = a)$ and is specified as

$$P^0 = \begin{pmatrix} 0.85 & 0.15 \\ 0.25 & 0.75 \end{pmatrix}, P^1 = \begin{pmatrix} 0.65 & 0.35 \\ 0.15 & 0.85 \end{pmatrix}.$$

The reward function is given by $R \in \mathbb{R}^{2 \times 2}$ with entries $R_{a,s} = r(s, a)$:

$$R = \begin{pmatrix} 1 & 0 \\ 0 & 1 \end{pmatrix}$$

The observation kernel $\beta \in \mathbb{R}^{2 \times 3}$ has entries $\beta_{s,o} = \mathbb{P}(o_t = o | s_t = s)$:

$$\beta = \begin{pmatrix} 0.80 & 0.10 & 0.10 \\ 0.30 & 0.65 & 0.05 \end{pmatrix}.$$

We use discount factor $\gamma = 0.9$.

In both figures, the blue lines are samples from the infinite family of piecewise linear inequalities given in Theorem 3.9,

$$y^\top \left( A(p^c)x - b(p^c) \right) \leq \sum_{(o,a)} p^\Delta_{(o,a)} \left| y^\top (A^{(o,a)}x - b^{(o,a)}) \right|,$$

indexed by $y \in \mathbb{R}^{|\mathcal{S}|}$. We sample 50 unit vectors $y$ uniformly on $S^1$ and plot the resulting boundaries. The intersection over all $y \in \mathbb{R}^{|\mathcal{S}|}$ is both necessary and sufficient.

The red and green dashed curves are the boundary curves of the semi-algebraic description from Theorem 4.2, obtained as the level sets $\{x : |q_j^{(i)}(x)| = 1\}$ for $i \in \{1, 2\}$ and $j$ ranging over the components of $q^{(i)}$. We compute these by solving the linear systems $D_1(x) q^{(1)} = C_1(x)$ and $D_2(x) q^{(2)} = C_2(x)$ for $q^{(1)}(x), q^{(2)}(x)$ at each $x$, and tracing the curves where any component reaches $\pm 1$.

For the boundary computation to be exact, $D_1, D_2$ must be invertible. From the parametrizations in eqs. 5–7, invertibility of $D_1, D_2$ reduces to invertibility of $A^{(o,a)}, B^{(o,a)}$, which holds generically: $A^{(o,a)}$ is a scaled transition-kernel block, and $B^{(o,a)}$ is a shifted version. When $D_1, D_2$ are non-square, $q^{(1)}, q^{(2)}$ must be obtained by solving a feasibility problem with box constraints, for which one practical approach is to solve the least-squares problems $\min \|D_i q^{(i)} - C_i\|^2$. This can introduce false positives, but the empirical error is small.

The figure illustrates that the same set $\mathcal{V}$ requiring infinitely many linear inequalities under Theorem 3.9 can be exactly recovered using only four polynomial curves under Theorem 4.2. The four curves here are specific to the $(|\mathcal{S}|, |\mathcal{A}|, |\mathcal{O}|) = (2, 2, 3)$ instance; in general the number of scalar boundary constraints is $|\mathcal{O}||\mathcal{A}| + |\mathcal{O}|(|\mathcal{A}| - 1)$.

## E.2. Figure 4

Let the state space be $\mathcal{S} = \{0, 1\}$ and the action space $\mathcal{A} = \{0, 1\}$. For each action $a \in \mathcal{A}$, the transition kernel $P^a \in \mathbb{R}^{2 \times 2}$ has entries $P^a_{ss'} = \mathbb{P}(s_{t+1} = s' | s_t = s, a_t = a)$ and is defined as

$$P^0 = \begin{pmatrix} 0.85 & 0.15 \\ 0.25 & 0.75 \end{pmatrix}, \qquad P^1 = \begin{pmatrix} 0.65 & 0.35 \\ 0.15 & 0.85 \end{pmatrix}.$$

Rewards are state-dependent and action-dependent via $R \in \mathbb{R}^{2 \times 2}$ with entries $R_{a,s} = r(s, a)$:

$$R = \begin{pmatrix} 1 & 0 \\ 0 & 1 \end{pmatrix},$$

Observations take values in $\mathcal{O} = \{0, 1\}$ and are generated from an observation (emission) kernel $\beta \in \mathbb{R}^{2\times 2}$ with entries $\beta_{s,o} = \mathbb{P}(o_t = o | s_t = s)$:

$$\beta = \begin{pmatrix} 0.65 & 0.35 \\ 0.35 & 0.65 \end{pmatrix}.$$

We consider a set of initial state distributions $\rho$ of the form $\rho = (\rho_0, \rho_1)$ with $\rho_0 + \rho_1 = 1$. In the figure we evaluate several representative initial state distributions spanning the simplex:

$$\rho^{(1)} = (0.2, 0.8), \quad \rho^{(2)} = (0.5, 0.5), \quad \rho^{(4)} = (0.75, 0.25), \quad \rho^{(5)} = (0.8, 0.2).$$

For each $\rho^{(i)}$, we maximize the scalar objective via vanilla gradient ascent with a fixed learning rate.

$$J(\pi; \rho^{(i)}) = \sum_{s \in \mathcal{S}} \rho_s^{(i)} V^\pi(s).$$

### E.3. Finite Description for $2 \times 2$ Real Parametric Systems

Consider a $2 \times 2$ real parametric system in 2 parameters with

$$A(p) = A^0 + p_1 A^1 + p_2 A^2$$
$$b(p) = b^0 + p_1 b^1 + p_2 b^2.$$

Let

$$A^0 = \begin{pmatrix} a_{11}^0 & a_{12}^0 \\ a_{21}^0 & a_{22}^0 \end{pmatrix}, \quad b^0 = \begin{pmatrix} b_1^0 \\ b_2^0 \end{pmatrix},$$

and for $k = 1, 2$,

$$A^k = \begin{pmatrix} a_{11}^k & a_{12}^k \\ a_{21}^k & a_{22}^k \end{pmatrix}, b^k = \begin{pmatrix} b_1^k \\ b_2^k \end{pmatrix}, p_k^\Delta = \Delta_k, p^c = 0.$$

For $x = (x_1, x_2)$, the midpoint residual vector is

$$C(x) = \begin{pmatrix} r_1(x) \\ r_2(x) \end{pmatrix}$$
$$= \begin{pmatrix} a_{11}^0 x_1 + a_{12}^0 x_2 - b_1^0 \\ a_{21}^0 x_1 + a_{22}^0 x_2 - b_2^0 \end{pmatrix}.$$

The matrix $D(x) = \begin{bmatrix} d^1(x) & d^2(x) \end{bmatrix} \in \mathbb{R}^{2\times 2}$ has columns

$$d^k(x) = \Delta_k (A^k x - b^k)$$
$$= \Delta_k \begin{pmatrix} a_{11}^k x_1 + a_{12}^k x_2 - b_1^k \\ a_{21}^k x_1 + a_{22}^k x_2 - b_2^k \end{pmatrix}, \quad k = 1, 2.$$

We have that $(I - D D^\dagger)C = 0$ iff the augmented matrix $[D|C]$ has the same rank as $D$. The latter is the case iff the $(r + 1) \times (r + 1)$ minors of $[D|C]$ vanish, where $r$ is the rank of $D$.

### E.4. Illustration of Theorem 4.4

Since the conditions in Theorem 4.4 involve taking determinants of the $B$-submatrices of $C(x)$, it is interesting to consider how those determinants factor as variables in $x$.

The process is as follows: after constructing quantities $M(x) \in \mathbb{R}^{n\times k}, R \in \mathbb{R}^{r\times k}, C(x) \in \mathbb{R}^{n+r\times k}, f(x) \in \mathbb{R}^{n+r}$, we take all $\binom{k}{l}$ bases formed by subsets of $\{1, \ldots, k\}$ of size $\ell$. For each basis $B$ we compute $C_B(x)$, the $\ell \times \ell$ submatrices of $C(x)$ indexed by $B$.

For each $t = 1, \ldots, \ell$, we calculate the sign polynomial $\det C_{B,t}(x) \cdot \det C_B(x)$, which is polynomial in $x$. We have via Theorem 4.4 that $x$ is a feasible value function iff there exists a basis $B$ such that $\det C_B(x) \neq 0$ and the sign polynomials are all non-negative:

$$\det C_{B,1}(x) \cdot \det C_B(x) \geq 0$$
$$\det C_{B,2}(x) \cdot \det C_B(x) \geq 0$$
$$\cdots$$
$$\det C_{B,\ell}(x) \cdot \det C_B(x) \geq 0.$$

If $B$ is a valid basis in the sense above, then consider the component of the solution set $\mathcal{S}$ generated by $B$,

$$S_B = \{x : \det C_B(x) \neq 0\} \cap_{t=1}^{\ell} \{x : \det C_{B,t}(x) \det C_B(x) \geq 0\}.$$

The structure of $S_B$ is controlled by the factorization of the polynomials $\det C_{B,t}(x) \cdot \det C_B(x)$ over $\mathbb{R}[x]$. Each irreducible factor cuts out an algebraic hypersurface in $\mathbb{R}^n$, and $\partial S_B$ is a union of pieces of these hypersurfaces. When all irreducible factors are linear in $x$, the component $S_B$ is a polyhedron.

To illustrate, we work out the factorization explicitly for the minimal nontrivial instance $|S| = |A| = |O| = 2$, where $k = rm = 4$ and $\ell = n + r = 4$, so there is a unique basis $B = \{1, 2, 3, 4\}$. Introduce the auxiliary polynomials

$$Q_s(x) := \left[ \gamma \sum_{s'} \alpha(s'|s, 0) \, x_{s'} + r(s, 0) \right] - \left[ \gamma \sum_{s'} \alpha(s'|s, 1) \, x_{s'} + r(s, 1) \right],$$
$$B_{s,a}(x) := \gamma \sum_{s'} \alpha(s'|s, a) \, x_{s'} + r(s, a) - x_s.$$

We refer to $Q_s$ as the advantage at state $s$ and to $B_{s,a}$ as the residual at $(s, a)$. Both are linear in $x$ and independent of $\beta$.

Direct computation in Macaulay2 (Grayson & Stillman) yields the following: for a generic POMDP, each sign condition $\det C_{B,1}(x) \cdot \det C_B(x)$ factors into two linear advantage factors and a polynomial whose coefficients depend on $\beta$ which is not necessarily linear:

$$\det C_{B,1}(x) \cdot \det C_B(x) = Q_0(x) \, Q_1(x) \cdot K_t(x; \, \alpha, r, \gamma, \beta).$$

In the fully observable case ($\beta = I$), however, each sign polynomial factors completely into linear terms:

$$\det C_{B,1}(x) \cdot \det C_B(x) = -Q_0(x) Q_1(x) \left[ Q_1(x) B_{0,1}(x) \right],$$
$$\det C_{B,2}(x) \cdot \det C_B(x) = \phantom{-} Q_0(x) Q_1(x) \left[ Q_1(x) B_{0,0}(x) \right],$$
$$\det C_{B,3}(x) \cdot \det C_B(x) = -Q_0(x) Q_1(x) \left[ Q_0(x) B_{1,1}(x) \right],$$
$$\det C_{B,4}(x) \cdot \det C_B(x) = \phantom{-} Q_0(x) Q_1(x) \left[ Q_0(x) B_{1,0}(x) \right].$$

The corresponding component $S_B$ is therefore cut out by sign conditions on a finite collection of hyperplanes—the advantage indifference planes $\{Q_s = 0\}$ and the Bellman-tight planes $\{B_{s,a} = 0\}$—recovering the MDP value-function polytope of (Dadashi et al., 2019).

In the partially observable case, the $K_t$ factor no longer splits — for generic $\beta$, $K_t$ is an irreducible quadratic in $x$ whose coefficients are polynomially dependent on the entries of $\beta$ and on $\alpha, r, \gamma$.

The zero locus $\{K_t = 0\}$ is therefore a conic in $\mathbb{R}^n$, not a union of two lines, and the boundary $\partial S_B$ acquires curved pieces.

Figure 5 displays the resulting picture for a representative $2 \times 2 \times 2$ instance. On the left, with $\beta = I$, the feasible set $\mathcal{V}$ is the quadrilateral spanned by the four deterministic policies, and its edges lie along the Bellman-residual lines $\{B_{s,a} = 0\}$. On the right, the deterministic policies still lie on $\partial \mathcal{V}$ but no longer span a polytope, and meanwhile portions of $\partial \mathcal{V}$ are traced out by the conics $\{K_t = 0\}$, in particular the concave arc on the lower-left edge of the feasible set.

For the instance plotted in Figure 5, we take

$$\alpha(\cdot|0, 0) = (\tfrac{4}{5}, \tfrac{1}{5}), \quad \alpha(\cdot|0, 1) = (\tfrac{2}{5}, \tfrac{3}{5}), \quad \alpha(\cdot|1, 0) = (\tfrac{3}{10}, \tfrac{7}{10}), \quad \alpha(\cdot|1, 1) = (\tfrac{9}{10}, \tfrac{1}{10}),$$

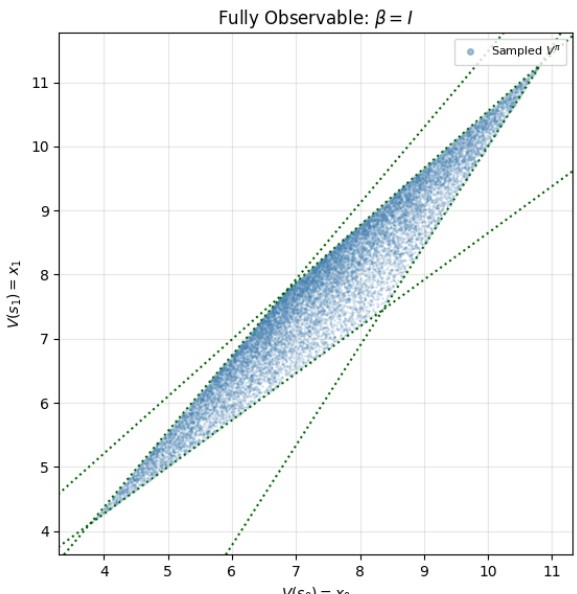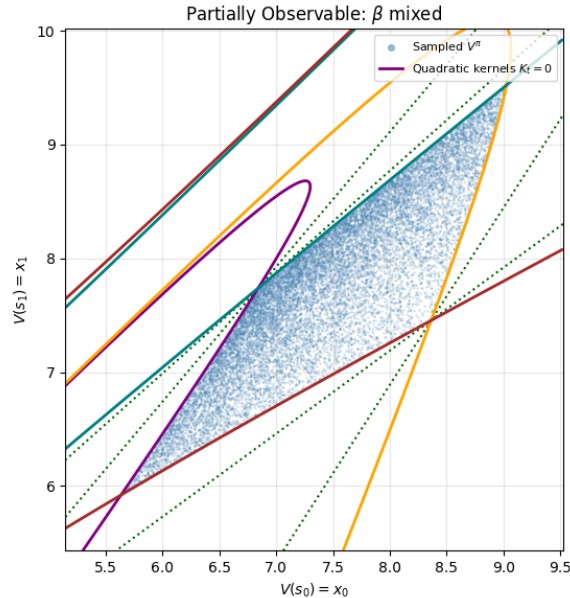

*Figure 5.* We construct the feasible set of value functions for a 2 state, 2 action, 2 observation MDP in a fully observable and partially observable setting. On the left panel, the calculation of $S_B$ generates a system of polynomials which all factor into linear terms. These linear functions are plotted in green dashes. On the right panel, the calculation of $S_B$ generates a system of polynomials which do not all factor into linear terms — the linear terms which survive for the fully observable case are plotted in green, while the irreducible non-linear terms are plotted in gold and yellow.

$$r(0,0) = 1,\ r(0,1) = \tfrac{1}{5},\ r(1,0) = \tfrac{1}{2},\ r(1,1) = \tfrac{3}{2},\ \gamma = \tfrac{9}{10}.$$

The four $\beta$-independent advantage and Bellman-residual polynomials are

$$Q_0(x) = \tfrac{9}{25}x_0 - \tfrac{9}{25}x_1 + \tfrac{4}{5}, \qquad\qquad Q_1(x) = -\tfrac{27}{50}x_0 + \tfrac{27}{50}x_1 - 1,$$
$$B_{0,0}(x) = -\tfrac{7}{25}x_0 + \tfrac{9}{50}x_1 + 1, \qquad\qquad B_{0,1}(x) = -\tfrac{16}{25}x_0 + \tfrac{27}{50}x_1 + \tfrac{1}{5},$$
$$B_{1,0}(x) = \tfrac{27}{100}x_0 - \tfrac{37}{100}x_1 + \tfrac{1}{2}, \qquad\qquad B_{1,1}(x) = \tfrac{81}{100}x_0 - \tfrac{91}{100}x_1 + \tfrac{3}{2}.$$

**Fully observable case ($\beta = I$).** Each kernel $K_t$ factors into two linear pieces:

$$K_0(x) = -\tfrac{1}{2500}\,(27x_0 - 27x_1 + 50)(32x_0 - 27x_1 - 10),$$
$$K_1(x) = \phantom{-}\tfrac{1}{2500}\,(14x_0 - 9x_1 - 50)(27x_0 - 27x_1 + 50),$$
$$K_2(x) = -\tfrac{1}{2500}\,(9x_0 - 9x_1 + 20)(81x_0 - 91x_1 + 150),$$
$$K_3(x) = \phantom{-}\tfrac{1}{2500}\,(9x_0 - 9x_1 + 20)(27x_0 - 37x_1 + 50).$$

Each linear factor coincides (up to a rational scalar) with one of the $Q_s$ or $B_{s,a}$ polynomials above; for example, $32x_0 - 27x_1 - 10 = 50\,B_{0,1}(x)$ and $27x_0 - 37x_1 + 50 = 100\,B_{1,0}(x)$. Together with the prefactor $Q_0Q_1$ that was divided out, each sign polynomial $S_{B,t} = Q_0Q_1K_t$ is therefore a product of four linears, and the feasible component $S_B$ is cut out entirely by hyperplane sign conditions, producing the polytope on the left panel.

**Partially observable case ($\beta = \left(\begin{smallmatrix} 7/10 & 3/10 \\ 1/5 & 4/5 \end{smallmatrix}\right)$).** The kernels become irreducible conics over $\mathbb{Q}$:

$$K_0(x) = -\tfrac{1}{10000}\big(945x_0^2 - 1620x_0x_1 + 675x_1^2 + 346x_0 + 174x_1 - 2600\big),$$
$$K_1(x) = \phantom{-}\tfrac{1}{10000}\big(459x_0^2 - 648x_0x_1 + 189x_1^2 - 1634x_0 + 2154x_1 - 4600\big),$$
$$K_2(x) = -\tfrac{1}{10000}\big(675x_0^2 - 1530x_0x_1 + 855x_1^2 - 3626x_0 + 4006x_1 - 4400\big),$$
$$K_3(x) = \phantom{-}\tfrac{1}{10000}\big(189x_0^2 - 558x_0x_1 + 369x_1^2 + 1646x_0 - 2026x_1 + 2400\big).$$

The discriminant of the quadratic form $945x_0^2 - 1620x_0x_1 + 675x_1^2$ in $K_0$ is $1620^2 - 4 \cdot 945 \cdot 675 = 71{,}100 > 0$, so the conic $\{K_0 = 0\}$ is a hyperbola; the analogous check for $K_1, K_2, K_3$ gives positive discriminants in every case, so all four kernels are hyperbolas. Their arcs trace the curved portions of $\partial \mathcal{V}$ visible in the right panel of Figure 5.

# F. Details on Experiments

Code for generating the experiments described below can be obtained at:

https://github.com/ryan-a-anderson/pomdp-value-geometry.

All experiments draw POMDP instances from the following distribution. Fix a triple $(S, A, O)$ and discount $\gamma = 0.9$. Sample:

- **Transition kernels.** For each action $a \in \{0, \ldots, A - 1\}$ and each state $s$, draw the row $\alpha_a(\cdot|s) \sim \mathrm{Dirichlet}(\mathbf{1}_S)$. This yields an $S \times S$ row-stochastic matrix per action, uniform over the simplex.

- **Observation kernel.** For each state $s$, draw $\beta(\cdot|s) \sim \mathrm{Dirichlet}(\mathbf{1}_O)$, yielding an $S \times O$ row-stochastic matrix.

- **Rewards.** For each action $a$ and state $s$, draw $r_a(s) \sim \mathrm{Uniform}[0, 10]$ independently.

- **Initial state distribution.** Unless otherwise noted, draw $\rho \sim \mathrm{Dirichlet}(\mathbf{1}_S)$.

All randomness is driven by a single `numpy.random.default_rng` seeded to `42` for reproducibility.

Given a partially observable instance $\langle S, A, O, \alpha, \beta, r, \gamma \rangle$, we construct the fully observable baseline $\langle S, A, \alpha, r, \gamma \rangle$ by replacing the observation kernel with the identity (so that $O = S$ and $\beta = I_S$), while holding $\alpha$, $r$, $\gamma$, and $\rho$ fixed. This isolates the effect of partial observability on the optimization landscape, controlling for the particular random dynamics and rewards sampled for each instance.

Memoryless stochastic policies are parametrized by softmax on observation-conditioned logits. A logit matrix $L \in \mathbb{R}^{O \times A}$ induces the policy

$$\pi(a|o) \;=\; \frac{\exp(L_{o,a})}{\sum_{a'} \exp(L_{o,a'})}.$$

For the fully observable baseline the same construction is used with $O = S$.

Given $\pi$, the state-to-state Markov chain under $\pi$ has transition matrix

$$P_{s,s'}^\pi \;=\; \sum_a \tau_{s,a} \, \alpha_a(s'|s), \qquad \tau_{s,a} \;=\; \sum_o \beta(o|s) \, \pi(a|o),$$

and per-state expected reward $r_s^\pi = \sum_a \tau_{s,a} \, r_a(s)$. The value vector is obtained by directly solving the Bellman linear system

$$(I - \gamma P^\pi) \, V^\pi \;=\; r^\pi$$

with `numpy.linalg.solve`. The scalar objective is $J(\pi) = \rho^\top V^\pi$.

We use vanilla gradient ascent with a fixed learning rate. The update is

$$L \;\leftarrow\; L + \eta \, \nabla_L J(\pi_L),$$

with learning rate $\eta = 0.005$ and $T = 3000$ steps. Logits are initialized as independent standard normals $L_{o,a} \sim \mathcal{N}(0, 1)$.

## F.1. Spread of Policy-Gradient Outcomes Under Partial vs. Full Observability

We test the curved semi-algebraic characterization of the set of feasible value functions as follows: on partially observable instances, vanilla policy gradient from independent random initializations should converge to a materially spread-out distribution of outcomes in both value and policy space, while on matched fully observable instances the same procedure should collapse to a single outcome up to numerical precision.

For each configuration $(S, A, O)$ on the grid

$$(S, A, O) \;\in\; \{4, 8, 12\} \times \{2, 3, 4\} \times \{2, 3\},$$

we sample random POMDP instances as described in §1.1. For each instance:

1. Draw an initial state distribution $\rho \sim \mathrm{Dirichlet}(\mathbf{1}_S)$.

2. Run 50 independent optimization trajectories from i.i.d. standard-normal logit initializations, each with $\eta = 0.005$ and $T = 3000$ steps.

3. Record the terminal triple $(J_i, V_i, \pi_i)$ for $i = 1, \ldots, 50$.

4. Build the fully observable baseline and repeat steps 1–3, *with the same $\rho$*, on the baseline instance.

For a given instance we then compute three aggregate statistics over the 50 restarts:

- **Value spread** — the range $\max_i J_i - \min_i J_i$.

- **Suboptimal fraction** — the share of restarts $i$ for which $\max_j J_j - J_i > 0.01$.

- **Policy spread** — the average pairwise $\ell_1$ distance

$$\frac{1}{\binom{50}{2}} \sum_{i<j} \|\pi_i - \pi_j\|_1,$$

where $\pi_i \in \mathbb{R}^{O \times A}$ is regarded as a flat vector.

The per-instance statistics are then aggregated across instances within each configuration, reporting the mean and standard deviation of each. The table below reports aggregate statistics over instances for each configuration.

*Table 1.* Aggregate statistics for Experiment A across all 18 configurations.

| Config $(S, A, O)$ | Value Spread | | Subopt. Frac. | | Policy Spread | |
| --- | --- | --- | --- | --- | --- | --- |
| | Partial | Full | Partial | Full | Partial | Full |
| $(4, 2, 2)$ | $18.783 \pm 11.845$ | $0.007 \pm 0.006$ | $0.263 \pm 0.216$ | $0.000 \pm 0.000$ | $0.317 \pm 0.145$ | $0.009 \pm 0.021$ |
| $(4, 2, 3)$ | $17.062 \pm 16.408$ | $0.013 \pm 0.020$ | $0.103 \pm 0.205$ | $0.000 \pm 0.000$ | $0.117 \pm 0.194$ | $0.007 \pm 0.015$ |
| $(4, 3, 2)$ | $15.672 \pm 20.881$ | $0.029 \pm 0.049$ | $0.223 \pm 0.264$ | $0.000 \pm 0.000$ | $0.192 \pm 0.175$ | $0.006 \pm 0.013$ |
| $(4, 3, 3)$ | $23.090 \pm 8.862$ | $0.041 \pm 0.033$ | $0.287 \pm 0.218$ | $0.000 \pm 0.000$ | $0.279 \pm 0.167$ | $0.005 \pm 0.008$ |
| $(4, 4, 2)$ | $5.704 \pm 9.200$ | $0.270 \pm 0.422$ | $0.160 \pm 0.228$ | $0.003 \pm 0.008$ | $0.131 \pm 0.168$ | $0.016 \pm 0.031$ |
| $(4, 4, 3)$ | $24.628 \pm 18.339$ | $0.301 \pm 0.229$ | $0.180 \pm 0.219$ | $0.000 \pm 0.000$ | $0.144 \pm 0.140$ | $0.011 \pm 0.011$ |
| $(8, 2, 2)$ | $26.695 \pm 4.932$ | $0.043 \pm 0.051$ | $0.220 \pm 0.145$ | $0.000 \pm 0.000$ | $0.314 \pm 0.153$ | $0.006 \pm 0.008$ |
| $(8, 2, 3)$ | $18.780 \pm 14.591$ | $0.028 \pm 0.018$ | $0.120 \pm 0.125$ | $0.000 \pm 0.000$ | $0.184 \pm 0.173$ | $0.011 \pm 0.014$ |
| $(8, 3, 2)$ | $19.638 \pm 11.285$ | $0.054 \pm 0.057$ | $0.313 \pm 0.182$ | $0.000 \pm 0.000$ | $0.312 \pm 0.088$ | $0.020 \pm 0.036$ |
| $(8, 3, 3)$ | $25.778 \pm 14.546$ | $0.158 \pm 0.120$ | $0.220 \pm 0.192$ | $0.000 \pm 0.000$ | $0.233 \pm 0.122$ | $0.030 \pm 0.031$ |
| $(8, 4, 2)$ | $25.460 \pm 18.234$ | $0.180 \pm 0.302$ | $0.270 \pm 0.200$ | $0.003 \pm 0.008$ | $0.237 \pm 0.128$ | $0.019 \pm 0.037$ |
| $(8, 4, 3)$ | $10.616 \pm 7.909$ | $0.204 \pm 0.279$ | $0.293 \pm 0.220$ | $0.050 \pm 0.122$ | $0.240 \pm 0.148$ | $0.054 \pm 0.058$ |
| $(12, 2, 2)$ | $15.348 \pm 8.547$ | $0.041 \pm 0.037$ | $0.197 \pm 0.159$ | $0.000 \pm 0.000$ | $0.279 \pm 0.155$ | $0.010 \pm 0.019$ |
| $(12, 2, 3)$ | $14.260 \pm 5.207$ | $0.051 \pm 0.039$ | $0.243 \pm 0.166$ | $0.000 \pm 0.000$ | $0.328 \pm 0.185$ | $0.009 \pm 0.014$ |
| $(12, 3, 2)$ | $19.506 \pm 11.628$ | $0.128 \pm 0.131$ | $0.293 \pm 0.187$ | $0.000 \pm 0.000$ | $0.297 \pm 0.117$ | $0.024 \pm 0.033$ |
| $(12, 3, 3)$ | $19.336 \pm 9.808$ | $0.105 \pm 0.114$ | $0.223 \pm 0.180$ | $0.000 \pm 0.000$ | $0.243 \pm 0.127$ | $0.039 \pm 0.041$ |
| $(12, 4, 2)$ | $13.337 \pm 9.849$ | $0.202 \pm 0.403$ | $0.367 \pm 0.177$ | $0.003 \pm 0.008$ | $0.298 \pm 0.082$ | $0.002 \pm 0.004$ |
| $(12, 4, 3)$ | $18.199 \pm 6.600$ | $0.151 \pm 0.171$ | $0.267 \pm 0.149$ | $0.000 \pm 0.000$ | $0.254 \pm 0.119$ | $0.019 \pm 0.025$ |

Across all 18 configurations, the partial-observability value spread exceeds the fully observable value spread by at least an order of magnitude (typically two), and the suboptimal fraction under partial observability is in the range 0.1–0.37, compared with 0.00–0.05 in the fully observable baseline.

The qualitative gap is consistent with the theoretical prediction. In the fully observable case the feasible value set is a finite union of polytopes (Dadashi et al., 2019), and the only stationary points of a linear objective restricted to that set are vertices; vanilla gradient ascent with generic initializations reaches one of those vertices with probability one and the distribution over restarts collapses. Under partial observability, Theorem 4.2 and Corollary 4.3 identify a semi-algebraic feasible set whose boundary carries curved (higher-degree) components.

## F.2. Counting Distinct Local Optima

The experiment in Appendix F.1 documents that the distribution of outcomes under random restarts is spread out, but does not directly count distinct local optima. The present experiment is designed to estimate how many local optima are encountered in practice, and how this number scales with the size $S$ of the state space.

For each configuration on the grid $S \in \{8, 12, 16, 20, 24\}$, $A \in \{2, 3, 4\}$, $O \in \{2, 3\}$:

1. Sample $n_{\mathrm{inst}} = 20$ independent POMDP instances.

2. For each instance, sample $n_\rho = 10$ independent initial state distributions $\rho \sim \mathrm{Dirichlet}(\mathbf{1}_S)$.

3. For each (instance, $\rho$) pair, run $n_{\mathrm{starts}} = 50$ independent optimization trajectories of length $T = 3000$ at $\eta = 0.005$ from i.i.d. standard-normal logit initializations.

4. Cluster the 50 terminal values $\{J_i\}$ with tolerance $\delta_J = 0.1$. The number of clusters is reported as the estimated number of distinct local optima for that (instance, $\rho$) pair.

Pooling over $A$, $O$, instances, and $\rho$, for each state-space size $S$ we then report:

- Mean number of optima, averaged over all (instance, $\rho$) pairs with that $S$.

- Share with more than one optimum, i.e. the fraction of pairs whose cluster count exceeds 1.

- Mean $\|V\|$-distance between best and worst cluster, conditional on there being more than one cluster.

To estimate the number of distinct local optima reached, the converged objective values $\{J_i\}_{i=1}^N$ across random restarts are clustered by a simple single-pass procedure sorted by descending $J$: a new cluster is opened whenever a run's $J$-value differs from every existing cluster mean by more than a tolerance $\delta_J$. For the reported runs $\delta_J = 0.1$. The suboptimal fraction is defined as the share of restarts *not* in the cluster of highest mean $J$, and the reported "$\|V\|$ distance" is the Euclidean distance in $\mathbb{R}^S$ between representative value vectors from the best and worst cluster.

Table 2. Experiment B results pooled by state-space size $S$.

| $S$ | Mean # optima | Share with $>1$ optimum | Mean $\|V\|$-distance (best–worst) |
|---|---|---|---|
| 8 | 1.82 | 0.64 | 10.07 |
| 12 | 2.05 | 0.75 | 9.78 |
| 16 | 2.25 | 0.79 | 10.92 |
| 20 | 2.67 | 0.88 | 11.53 |
| 24 | 2.80 | 0.86 | 11.87 |

In the finer-grained table, $(8, 4, 3)$ already records a mean of 6.12 optima per (instance, $\rho$) with a maximum of 8; every instance in every configuration at $S \geq 8$, $A \geq 3$ exhibits at least two distinct optima.

Two features of the table are notable. First, both the frequency and the count of distinct optima grow monotonically with $S$: as the state space enlarges, the feasible value set acquires more boundary components capable of supporting isolated maximizers, exactly as the semi-algebraic characterization predicts. Second, the best-to-worst $\|V\|$-distance remains substantial across all $S$: the multi-optima phenomenon is not a numerical artifact, but reflects genuine separations of order 10 in value space.

A caveat worth stating: the reported optima counts are *lower bounds* on the true number of critical points of $J$ on the feasible set, because gradient ascent from 50 random inits need not visit every basin and the clustering tolerance $\delta_J = 0.1$ coalesces nearby optima. The monotone increase with $S$ should therefore be read as a lower-bound trend.

### F.3. Finite-Memory Policies via Observation Enrichment

We wanted to also consider whether the multi-optima phenomenon is an artifact of the memoryless policy class rather than a consequence of partial observability. The present experiment addresses this directly by running the Experiment A protocol on policy classes with finite memory $k$, implemented via observation enrichment, and tracking how value spread varies with $k$.

Following the perspective of Azizzadenesheli et al. (2020), a length-$k$ observation-history policy in the original POMDP is realized as a memoryless policy in an enriched POMDP whose observation space is the $k$-fold product of observation histories. Concretely, for memory length $k$ we define the enriched observation space $O^{(k)}$ and an enriched observation kernel $\beta^{(k)}$ such that memoryless policies $\pi\colon O^{(k)} \to \Delta^A$ correspond to length-$k$ policies in the original POMDP; the underlying state dynamics $\alpha$ and reward $r$ are unchanged.

Because the enriched policy class has dimension $|O^{(k)}| \cdot A$, which grows with $k$, the optimization budget is scaled with $k$ to control for optimization difficulty (as opposed to landscape difficulty).

Below, we run the experiment on the small grid $(S, A, O) \in \{(4, 2, 2), (4, 3, 2), (8, 2, 2), (8, 3, 2)\}$ for instances:

- $k = 0$: memoryless policies (the original class).

- $k = 1$: length-1 observation-history policies.

- $k = 2$: length-2 observation-history policies.

- Full: the fully observable baseline

All four share the same underlying $\alpha$, $r$, $\gamma$, and $\rho$.

*Table 3.* Experiment C results for $k \in \{0, 1\}$ (continued in Table 4).

| Config $(S, A, O)$ | $k = 0$ | | $k = 1$ | |
|---|---|---|---|---|
| | Spread | Subopt. | Spread | Subopt. |
| $(4, 2, 2)$ | $1.126 \pm 2.407$ | $0.315 \pm 0.416$ | $1.122 \pm 2.410$ | $0.110 \pm 0.232$ |
| $(4, 3, 2)$ | $1.339 \pm 2.355$ | $0.030 \pm 0.054$ | $0.243 \pm 0.368$ | $0.215 \pm 0.427$ |
| $(8, 2, 2)$ | $0.284 \pm 0.385$ | $0.205 \pm 0.303$ | $0.523 \pm 0.719$ | $0.260 \pm 0.369$ |
| $(8, 3, 2)$ | $0.287 \pm 0.438$ | $0.255 \pm 0.423$ | $0.305 \pm 0.513$ | $0.275 \pm 0.418$ |

*Table 4.* Experiment C results for $k = 2$ and the fully observable baseline (continued from Table 3).

| Config $(S, A, O)$ | $k = 2$ | | Full | |
|---|---|---|---|---|
| | Spread | Subopt. | Spread | Subopt. |
| $(4, 2, 2)$ | $0.111 \pm 0.239$ | $0.125 \pm 0.280$ | $0.021 \pm 0.022$ | $0.000 \pm 0.000$ |
| $(4, 3, 2)$ | $0.292 \pm 0.409$ | $0.125 \pm 0.240$ | $0.055 \pm 0.089$ | $0.010 \pm 0.022$ |
| $(8, 2, 2)$ | $0.545 \pm 0.819$ | $0.310 \pm 0.394$ | $0.031 \pm 0.016$ | $0.000 \pm 0.000$ |
| $(8, 3, 2)$ | $0.138 \pm 0.202$ | $0.330 \pm 0.401$ | $0.113 \pm 0.036$ | $0.045 \pm 0.087$ |

Three patterns are consistent across configurations:

1. Increasing $k$ tends to reduce the value spread, at least for the smallest configurations where memory is enough to substantially disambiguate the state.

2. Partial observability still induces significant spread even at $k = 2$: the spread for $k = 2$ is consistently larger than the spread in the fully observable setting, which is essentially zero.

3. The suboptimal fraction does not cleanly shrink with $k$: on $(8, 2, 2)$ and $(8, 3, 2)$ it is non-monotone in $k$. This is consistent with the geometric reading: enlarging the policy class adds new feasible value functions and new critical points, which can shift the basin structure in ways that are not simply monotone in optimization-hardness indices.

Enlarging memory mitigates, but does not erase, the geometric complexity of the feasible value set.

