# OpenReview forum: "The Value Function Semi-Algebraic Set in Partially Observable Markov Decision Processes"
_ICML.cc/2026/Conference — ICML 2026 regular_

### Official Review · Reviewer_t8La · 2026-03-05

**Soundness:** 3
**Presentation:** 2
**Significance:** 2
**Originality:** 2
**Overall Recommendation:** 4
**Confidence:** 2

**Summary:**

This paper considers the model of infinite horizon POMDP problems with reactive (memoryless) policies. The main goal of this paper is to characterize the feasibility set (points with satisfy the Bellman policy evaluation equation) of the value function in POMDPs, which describes the different values that a reactive policy can have across all possible choices of such a policy. They provide theoretical results for the POMDP case by extending existing results on the fully observable MDP case. In particular, they use established theoretical results from interval and parametric matrix systems on MDPs. They express the overall parametric systems as the intersection of two systems and obtain the solution of the combined set of inequalities to describe the feasible space of the POMDP. Finally the provide a way to compute the feasible space with a finite number of inequalities and illustrate these results on a small toy POMDP problem.

**Compliance With Llm Reviewing Policy:**

Affirmed.

**Final Justification:**

The rebuttal addressed my concern to an extent. In addition, I'm changing my review to a weak accept to arrive at a consensus with the other reviewers.

**Key Questions For Authors:**

Please see my main comments 2,3,6.

**Limitations:**

yes

**Strengths And Weaknesses:**

The main claims and theoretical results seem to be correct. I found some presentation issues in terms of clearly specifying the used notation in the paper that I have mentioned in the "minor comments" which made an otherwise clearly written paper slightly harder to read. I found the main technical results interesting, but am not quite sure about how significant they are particularly because there is hardly any discussion on how these results can be used in practice, or also how/whether we can use these results when larger POMDPs are considered, and is it practical to do so. In terms of originality, I do not see any particularly interesting new findings. For example, it is already well know that the optimal stationary stochastic policy solution in POMDPs depends on the initial state distribution. If I missed anything, the authors are encouraged to correct me.


main comments

1. line 92 col 2: This seems a bit of a non-standard definition for an MDP. I would divide it into fully observable and partially observable MDPs. Saying that $M$ is an MDP is strange because in an MDP we do not have an observation function $\beta$ and observations $O$, we would just have the state $s$.

2. Figure 2 provide a very interesting visualization that helps understand the overall theoretical result. To generate this visualization, are you just using randomly sampled inequalities which are the blue lines? How do you select the $y$ vectors ? I ask this because it is an infinite space, but you are only displaying finitely many lines (50 as you mentioned). You mention much later in the text that the reader should refer to appendix C, I think it would be better to mention it when they are introduced in the text or in the caption. And also, the appendix doesn't contain information about how the plots are generated, but rather the model parameters of the POMDP.

3. Figure 3 shows the interesting result of showing the feasible space with just 4 inequalities rather than the theoretical feasible space which has to consider infinitely many inequalities. Is this possible to construct only because Theorem 4.2 gives us linear constraints (thereby allow us to just consider the extremities)?

4. The notion of optimality is not clearly defined in Figure 4. There are optimal belief based policies and then there are optimal stationary reactive policies. While possible in some examples, it isn't necessary that the performance of the optimal stationary reactive policies would match the optimal belief based policies. From the context of the paper, I think that you are referring to optimal policies within the class of stationary reactive policies. Note that one could also consider the more general non-stationary reactive policies, however this makes the analysis much more complicated, particularly for the infinite horizon case.

5. There is no discussion on the scalability of this approach and if it can be applied to larger POMDPs or more practical problems. The main results are just illustrated over a small toy POMDP problem.

6. While the obtain theoretical results are impressive and a solid contribution, the paper does not discuss how we can make use of the obtained feasibility landscape in an effective manner. For the smaller POMDP problem, the insights are quite intuitive, but if we take the case of a larger POMDP problem: is there a way we can analyze the resulting landscape? Perhaps by projecting to a lower dimension which consists of the value functions averaged over their stationary distribution. If this is not possible, then why is it not possible? The lack of further discussion on these fundamental ideas diminishes the overall contribution of the paper. If possible, could the authors suggest some ways to utilize the main result of this paper in some more detail.



minor comments

1. It isn't clear what a feasible value function is on reading the introduction. Maybe a light description in the introduction would help (like saying that a feasible solution satisfies the Bellman equation). I think the first time it is properly defined is in Theorem 3.8.

2. In remark 3.4, it would be good to use some notation apart from r, which might get confused with reward of the MDP.

3. What is the |x| notation in eq (4) ? Is it elementwise absolute value?

4. What is I_{s,s'} in eq (6) ?  Is this the probability that s will transition to s' under the current policy? Shouldn't there be a dependence on $\pi$ somewhere in eqs (6,7) because you are essentially re-representing eqs (1,2)?

5. You could have used C and d instead of B and c in Proposition 3.7 to avoid confusion in the notation of the two different systems.



typos

1. line 298 col 2: "deviations that appear" -> "deviations that appears"

2. line 395 col 1: "In Figure 3" -> "Figure 3"

---

> ### Author Rebuttal · Authors · 2026-03-31
>
> We thank the reviewer for this detailed and constructive reading.
>
> **1: Distribution-dependence of optimal policies**
>
> The distribution dependence of optimal memoryless policies is a known qualitative phenomenon (see, e.g., Puterman, 2014). We emphasize, however, that this is not the primary contribution of Section 5. Rather, Figure 4 illustrates how the geometry of the solution set provides a new explanation for this dependence—namely, the non-polyhedral, semi-algebraic structure of $\mathcal{V}$—as well as a framework for analyzing how many and which optimal policies arise for a given initial state distribution $\rho$.
>
> In contrast to the polyhedral setting, where every linear objective is optimized at a vertex, our semi-algebraic characterization shows that, in POMDPs, this dependence involves both combinatorial aspects (which boundary component is selected) and geometric aspects (where within that component the optimum lies).
>
> **2: Non-standard MDP definition**
>
> Our unified tuple $\langle S, A, O, \alpha, \beta, r, \gamma \rangle$ captures both settings by varying $\beta$ but may cause confusion as it allows for a partially observable setting from the outset. In the revision we will introduce the fully observable MDP as the standard tuple $\langle S, A, \alpha, r, \gamma \rangle$ first, then define the POMDP by augmenting with $O$ and $\beta$.
>
> **3: Sampling of the $y$-vectors in Figure 2**
>
> The $y$ vectors were sampled uniformly at random from the unit sphere $S^{|S|-1}$. Each sampled $y$ instantiates one piecewise-linear inequality from Theorem 3.8, and the 50 displayed lines are 50 such samples. We will add this explanation to the figure caption and add details in the appendix.
>
> **4: Why four curves suffice in Figure 3**
>
> Partially, yes, but the precise reason is the zonotope structure. The finite description of Theorem 4.2 requires $x$ to satisfy $D_1 q^{(1)} = C_1$ and $D_2 q^{(2)} = C_2$ for some $q^{(i)} \in [-1,1]^{|O||A|}$.
>
> The boundary of the feasible set corresponds to setting one component of some $q^{(i)}$ as $\pm 1$. For the 2-state, 2-action, 3-observation example in Figures 2 and 3, the parameter spaces for $q^{(1)}$ and $q^{(2)}$ are each 1-dimensional, so the constraints $|q^{(1)}| = 1$ and $|q^{(2)}| = 1$ each give two boundary curves (four total).
>
> In higher-dimensional examples the number of boundary pieces grows, but the finite description still holds: only finitely many polynomial equations and inequalities are needed.
>
> **5: Optimality notion in Figure 4**
>
> Figure 4 displays optimal policies within the class of memoryless stochastic policies; the caption will be updated to say so explicitly.
>
> **6: Scalability and practical utility**
>
> Our theoretical framework applies equally to large-scale POMDPs. A key practical implication is that it clarifies how POMDP parameters govern both the number and the degrees of the boundary components of the solution set. For optimization over a semi-algebraic set, the number of critical points depends on the number and degrees of its defining components (see, e.g., Draisma et al., The Euclidean Distance Degree, 2014).
>
> In the revised appendix, we provide explicit bounds on the number and degrees of boundary components for several classes of POMDPs, including state-aggregation models. Combining these bounds with algebraic complexity results for polynomial optimization yields predictions of the resulting optimization complexity in terms of the POMDP parameters (e.g., number of states, degree of partial observability). Beyond such predictions, our framework may also inform the design of methods based on convex relaxations and entropy regularization.
>
> **Minor comments**
>
> 1. Feasible value function – We will add a short definition in the introduction, e.g., "a value function is feasible if it satisfies the Bellman equation for some policy $\pi \in \Delta^O_A$," before the formal treatment in Theorem 3.8.
>
> 2. Notation $r$ in Remark~3.4 – We agree that the overuse of $r$ in referring to the sign pattern in the remark is potentially confusing, and we will change this to another notation.
>
> 3. $|x|$ in eq. (4). – Yes, $|x|$ denotes the componentwise absolute value. We will add a clarifying note when the notation is first introduced.
>
> 4. $I_{s,s'}$ in eq. (6). – $I_{s,s'}$ is the $(s,s')$ entry of the identity matrix (i.e., $\mathbf{1}[s = s']$), arising from the $I$ in $I - \gamma P^\pi$.  The policy dependence enters through the subsequent $A^{(o,a)}$ terms; eqs. (6)--(7) give the decomposition of the full coefficient matrix $A(p) = A^0 + \sum_{o,a} p_{o,a} A^{(o,a)}$, so $\pi$ enters through $p_{o,a}$.  We will add a clarifying sentence after eq. (7).
>
> 5. Notation $B, c$ in Proposition 3.7 – We agree that using $B, c$ for the second system risks confusion with generic matrix names. We will update the choice of matrix and vector names to reflect the fact that these are two similar systems, such as $A(p), b(p), \bar{A}(v), \bar{b}(v)$.

---

> > ### Author Rebuttal · Reviewer_t8La · 2026-04-03
> >
> > I thank the authors for addressing each of my points.
> >
> > 1. The precise contribution of this work in terms of the semi-algebraic characterization is more clear to me now.
> > 3. Thanks for the clarification, this is close to what I had in mind.
> > 4. Thank you for the detailed explanation, it makes more sense to me now.
> >
> > My main concern is still around point 6. While the level of theoretical contribution is good, it is also important to intuitively connect how these theoretical results are relevant in practice. In my opinion, Figure 4 is a step in the right direction to demonstrate the usefulness of the theory.
> >
> > A larger portion of the paper could have been focused on observing the effect of any standard policy gradient algorithms on such landscapes. Since policy gradient algorithms are notorious for converging to poor local minima, observing learning trajectories over such landscapes would have really helped illustrate why the exact problem occurs.
> >
> > For example, consider the bottom-right part Figure 4: I expect that if we initialize close to the upper vertex, then we will likely converge to the upper vertex because the gradients push the policy in that direction. When we get close to the upper vertex the policy gets stuck there because the gradients will never push it out of that point and towards the global optima. And so, it becomes quite important to think about initialization of the policy parameters, or develop a technique which doesn't greedily follow the local gradient.
> >
> > I'm not saying that this specific experiment be done, but any such simple demonstrations on even small - moderate sized POMDPs would have been really useful in highlighting the importance of the theoretical contributions. In addition, if we are considering larger POMDPs which cannot be visualized adequately, then how can we make use of the boundary data to study these landscapes? These topics were not discussed and would have been helpful in highlighting the usefulness of the theory.
> >
> > For these reasons, I am choosing option (c) and I would like to maintain my decision.

---

> > > ### Author Response · Authors · 2026-04-04
> > >
> > > We appreciate the reviewer’s encouragement to work on linking the theoretical results to empirical evaluation.
> > >
> > > The reviewer's intuition is correct regarding the fourth panel of Figure 4 – when a randomly initialized value function is close to a suboptimal extremum of the feasible set and the initial state distribution is aligned in such a way that the optimization updates converge to that suboptimal extremum, then the optimization trajectory will get stuck at that suboptimal extremum. This is a typical behavior of local or gradient based optimization algorithms in non-convex optimization, and this also happens in the partially observable setting since the domain is a semi-algebraic set with curved boundaries as shown in our theoretical results.
> > >
> > > As mentioned in the rebuttal, the theoretical description of the boundaries can be used to estimate the optimization complexity. We are cataloging the degrees and the corresponding predicted complexity for several families of POMDPs, depending on the degree of partial observability, and will add the detailed results in the revision.
> > >
> > > In order to provide empirical demonstration, we conducted several new experiments. Please see our Reply Rebuttal Comment to Reviewer eDuZ, which summarizes our systematic study across randomly generated POMDPs across differing state, observation, and action spaces. In particular, to quantify the multi-extremal structure of optimization in POMDPs, we ran an experiment where for each POMDP instance $(S,A,O,\alpha, \beta,r, \gamma)$, we sampled 25 initial state distributions $\rho$, and ran 100 random restarts of policy gradient over memoryless stochastic policies, then clustered the converged value functions. The number of clusters directly estimates the number of distinct local optima reached.
> > >
> > > The results in the table reveal that multiple distinct local optima are the norm rather than the exception. The fraction of (instance, $\rho$)-pairs exhibiting multiple local optima increases with the number of states as does the mean number of distinct local optima. These experiments confirm the pattern for a large number of configurations with dozens of states and confirm that the stuck trajectory behavior occurs in higher dimensions.
> > >
> > > Other experiments of interest include evaluation of the impact of the geometric structure on optimization for finite-memory policies. We believe that these empirical demonstrations provide a valuable illustration of the theoretical contributions. We hope the reviewer will agree that these results are helpful in highlighting the usefulness of the theory beyond toy models. Please let us know if there are any additional demonstrations or anything else you would like to see added during the response period. Thank you for your time and valuable feedback.

---

### Official Review · Reviewer_uGEk · 2026-03-11

**Soundness:** 3
**Presentation:** 3
**Significance:** 2
**Originality:** 4
**Overall Recommendation:** 5
**Confidence:** 3

**Summary:**

This paper provides a characterization of the feasible set of value functions in POMDPs as a semi-algebraic set. The set is defined via a list of inequalities based on the main objects building the POMDP.  This result extends Dadasi's work on MDPs, showing that the structure here is way more complex than a polytope: partial observability induces nonlinear constraints. The analysis reveals qualitative phenomena unique to POMDPs, including the emergence of isolated local maximizers of the long-term reward and their dependence on the initial state distribution.

**Compliance With Llm Reviewing Policy:**

Affirmed.

**Final Justification:**

Including the comments from the rebuttal will make the paper at acceptance level, as far as I am concerned.

**Key Questions For Authors:**

my main question is a fundamental doubt: I do understand that the analysis entailing non-markovian policies would be extremely more challenging, and I am not expecting a first contribution to cover it. BUT I see the need here to clarify one point: can the author clarify to what extent these results refer to the POMDP intrinsic structure and not to the (known) limits of markovian policies in POMDPs? Like pointing in which points the nature of the policy used have a crucial effect in the results. I believe it could be really helpful to add one hint against the claim "you characterize POMDPs with markovian policies when we know that non-markovian policies are needed", if even characterizing the problem for finite-memory policies turns out to be hard.

**Limitations:**

I was not able to find any argument of the limitations of the proposed method unfortunately.

**Strengths And Weaknesses:**

**Soundness**:
the analysis is thoughtful and sound,  visualizations are pretty clear in providing evidence for the claims. My only concern is reported in the questions.

**Presentation**:
The presentation is clear and I don't have further comments on this side.

**Significance**:
I believe, as for Dadashi's work, this contribution to go in the direction of helping understanding the fundamental properties of POMDPs, which is quite significant to me.

**Originality**:
the analysis employs tools which are quite original to me. The use of intervals matrix systems is intuitively similar in structure to the use of intervals for planning in POMDPs, but the results are way richer (I wonder if the same toolbox could be used for that as well).

---

> ### Author Rebuttal · Authors · 2026-03-31
>
> Thank you for your thoughtful review and comments. Your question regarding the intrinsic structure of POMDPs and the role of non-Markovian policies is an important conceptual point.
>
> In fully observable MDPs, there always exists an optimal policy that is both memoryless and stochastic. This property does not extend to POMDPs: policies with memory can outperform memoryless ones, and even within the memoryless class, stochastic policies can outperform deterministic policies. While the class of policies with memory is strictly richer, computing an optimal policy is PSPACE-complete in the finite-horizon setting and undecidable in the infinite-horizon setting. This motivates the study of optimization over the memoryless class as a tractable and theoretically meaningful proxy.
>
> Our results refer to the POMDP intrinsic structure. The nonlinear geometric constraints arise from partial observability itself, rather than from the restriction to memoryless policies, whereby there is a noteworthy interplay. Note that any limited-memory policy class can be recast as a class of Markovian policies via observation enrichment (see, e.g., Azizzadenesheli et al., 2020).
> Importantly, even after such augmentation, the system may remain partially observable, and the set of feasible value functions can still be subject to nonlinear constraints as described in our results, which in turn give rise to isolated local optima. Since our framework does not depend on the specific structure or cardinality of the observation space, the geometric insights we obtain for memoryless POMDPs extend to such enriched systems, and thus to such non-Markovian policy classes as well.
>
> We suggest that there is a natural trade-off. Namely, larger memory in principle can better identify the state, thus reducing partial observability and the degree of the constraints. The quantification of this tradeoff is a very interesting avenue for future investigation that we think can be naturally developed building on our presented theoretical results.
>
> Azizzadenesheli et al. (2020) https://arxiv.org/pdf/1810.07900

---

> > ### Author Rebuttal · Reviewer_uGEk · 2026-04-03
> >
> > All concerns properly addressed.

---

### Official Review · Reviewer_eofe · 2026-03-12

**Soundness:** 3
**Presentation:** 3
**Significance:** 3
**Originality:** 3
**Overall Recommendation:** 4
**Confidence:** 3

**Summary:**

It is know that the space of value functions for infinite-horizon discounted-reward tabular MDPs is a polytope, that is, the union of some convex polytopes. To be more precise, one may consider the set of vectors $(V^\pi(s))_{s\in S}$ for different policies $\pi$, then the set is a polytope. This paper extends this result to POMDPs. What is interesting is that the set of value functions for a POMDP is not necessarily a polytope while the paper proves that it is a semi-algebraic set. The proof is based on novel arguments by building parametric matrix systems for the Bellman equation and taking piecewise linear inequalities describing the set. Then, in Section 4, the paper shows that one may deduce a finite description, which turns out to be a semi-algebraic set.

**Compliance With Llm Reviewing Policy:**

Affirmed.

**Final Justification:**

The authors engaged constructively with the questions and comments and have addressed them.

**Key Questions For Authors:**

- Can we obtain the earlier result on MDPs as a corollary of the result of this paper?
- Can the authors explain possible strategies to design an efficient policy iteration algorithm based on the geometry result?

**Limitations:**

yes

**Strengths And Weaknesses:**

Strength
- The paper studies and settles the interesting question of extending the geometry result for the value functions of an MDP to POMDP.
- The result that we observe semi-algebraic sets for the case of POMDPs is interesting.
- Techniques for proving the result seem to be novel.

Weakness
- Its connections and implications to earlier works are not clearly demonstrated.
- As mentioned in the paper, the geometry result itself does not immediately imply efficient algorithms for policy optimization in POMDPs.

---

> ### Author Rebuttal · Authors · 2026-03-31
>
> **1: Can the MDP result be recovered as a corollary?**
>
> Yes, the MDP case arises as the special case in which the observation kernel $\beta$ is the identity (or, more generally, any deterministic injective map from states to observations). In this setting, the effective policy polytope $\Delta^A_{S,\beta}$ coincides with the full policy polytope $\Delta^A_S$. We make this correspondence explicit in the revised appendix.
>
> There, we present a quantifier-free description of the solution set of the Bellman equation for POMDPs, based on determinants of certain submatrices. The resulting polynomial conditions have degrees that reflect the level of observability in the POMDP. In particular, in the fully observable setting, all such determinantal factors are linear. This recovers the result of Dadashi et al. (2019), namely that the solution set for a fully observable MDP is a union of polytopes (cut out by linear constraints).
>
> **2: Strategies toward efficient policy iteration?**
>
> We outline two strategies for enhancing policy iteration algorithms which are better informed by the results obtained in our paper.
>
> a. Convex relaxation – Dressler et al. (2022) studied the reward optimization problem in memoryless POMDPs with deterministic observation kernels. In general, this is a polynomial optimization problem, and as such cannot be solved via convex approaches. They investigate the relaxation of the polynomial optimization problem into a semi-definite program via the Lasserre hierarchy and observe empirically that a first-order SDP relaxation is able to find critical points of the optimization problem with high probability given generic POMDP parameters. Our work provides a corresponding perspective for the value functions which could be used to characterize when convex relaxation is effective.
>
> b. Entropic regularization for policy iteration – Zamboni et al. (2024) studied the performance of entropy-regularized objectives for finding optimal policies in POMDPs and demonstrated the robustness of these objectives with respect to observation noise. Our results, especially as discussed in Section 5, provide further insight into why these entropy-regularized objectives can achieve better performance. In particular, the number and degree of the boundary components of the solution set to the Bellman equation for a POMDP will determine the number of sub-optimal isolated extreme value functions, as well as the probability of reaching those sub-optimal extrema given a randomly initialized state distribution. Since entropy-regularization will tend to pull an optimization trajectory away from these boundary components, it reduces the probability of reaching one of the sub-optimal extrema.
>
> **3: Note on Connections to Earlier Work**
>
> In regard to the reviewer's comments on connections to earlier works:  We will add a paragraph at the end of Section 1.2 (Related Works) explicitly mapping our framework to (i) the value function polytope of Dadashi et al. (2019), (ii) the rational parametrization results of Müller et al. (2022), and (iii) the robust MDP geometry of Wang et al. (2022), clarifying how our semi-algebraic result relates to each.
>
> Dadashi et al. (2019) https://proceedings.mlr.press/v97/dadashi19a.html
>
> Dressler et al. (2022) https://arxiv.org/pdf/2211.09439
>
> Zamboni et al. (2024) https://arxiv.org/abs/2406.02295
>
> Müller et al. (2022) https://arxiv.org/pdf/2110.07409
>
> Wang et al. (2022) https://arxiv.org/pdf/2201.12929

---

> > ### Author Rebuttal · Reviewer_eofe · 2026-04-03
> >
> > Thank you for explaining that the work of this paper covers the MDP case and for providing two possible approaches to develop policy iteration algorithms. That said, I will keep my positive score.

---

### Official Review · Reviewer_eDuZ · 2026-03-13

**Soundness:** 3
**Presentation:** 3
**Significance:** 3
**Originality:** 3
**Overall Recommendation:** 4
**Confidence:** 4

**Summary:**

**Summary**

The manuscript analyzes the geometry of feasible value functions in infinite-horizon Partially Observable Markov Decision Processes (POMDPs) restricted to memoryless stochastic policies. Dadashi et al. (2019) established that the feasible value function set for fully observable MDPs constitutes a polytope. Extending this foundational work, the authors prove that partial observability introduces fundamentally nonlinear constraints, forming a semi-algebraic set. The authors initially parameterize the Bellman equation as an interval linear system bounded by infinitely many piecewise linear inequalities. By applying zonotope properties and the Moore-Penrose pseudoinverse, they translate this infinite system into a finite geometric description. Consequently, this mathematical framework rationalizes why optimal POMDP policies vary with initial state distributions and exhibit isolated local maximizers.

**Compliance With Llm Reviewing Policy:**

Affirmed.

**Final Justification:**

Soundness (2 → 3). The rank-deficiency concern is fully resolved. The rank stratification argument, characterizing the column inclusion condition via minors of the augmented matrix [D|C] at each stratum with the finite union preserving semi-algebraic closure, is mathematically sound. The clarification that Theorem 4.2 and Corollary 4.3 do not depend on the pseudoinverse further strengthens the theoretical foundation. The (k choose l) complexity bound improves over the general doubly exponential worst case, though its scaling for large observation/action spaces warrants the promised detailed derivation.

Originality (3 → 3). The application of real algebraic geometry to parameterize the Bellman equation remains an original theoretical perspective. The authors successfully bridge the polyhedral structure of MDPs (Dadashi et al., 2019) and the semi-algebraic structure of POMDPs, recovering the MDP result as a corollary when the observation kernel is injective.

Significance (2 → 3). The second-round rebuttal substantially improved significance by addressing our two core concerns. First, the finite-memory experiments demonstrate that geometric pathologies persist under richer policy classes: increasing memory reduces value spread but partial observability still induces significant spread compared to the fully observable baseline, and suboptimal fractions remain nontrivial even at memory level k=2. This supports the claim that the pathologies are intrinsic to partial observability rather than artifacts of the memoryless restriction. Second, the systematic experiments across randomly generated POMDPs with up to 24 states show that multiple local optima are the norm (64-88% of instances exhibit multiple optima), with the mean number of distinct optima increasing with the number of states. These experiments directly demonstrate that standard policy gradient algorithms get trapped in the semi-algebraic landscape, establishing the empirical connection we requested. We note two anomalies in the finite-memory data: for (8,2,2), value spread increases from k=0 to k=2, and for (8,3,2), the suboptimal fraction at k=2 exceeds k=0. These deserve brief discussion in the revision, as they may reveal interesting phenomena about how memory interacts with the optimization landscape.

Presentation (3 → 3). The manuscript is logically structured with precise notation. The promised revisions (surfacing the rank stratification discussion, clarifying notation, adding experimental details) will improve accessibility.

Overall. The authors engaged constructively across two rounds with substantial new work. Our two core concerns from the first round, whether the geometric pathologies persist under richer policy classes and whether they manifest beyond toy models, are now largely addressed through new experiments with finite-memory policies and larger POMDPs. The belief-state extension remains future work, and the experiments use randomly generated POMDPs rather than standard benchmarks, but the evidence is now sufficient to support the paper's central claims. We adjust from 3 (weak reject) to 4 (weak accept).

**Key Questions For Authors:**

1. The explicit polynomial construction in Section 4.1 relies heavily on the Moore-Penrose pseudoinverse $D^+$. How does the theoretical framework mathematically resolve singular regions in the value space where the deviation matrix $D(x)$ becomes rank-deficient, given that the rank depends on $x$?
2. Quantifier elimination via the Tarski-Seidenberg theorem typically incurs extreme computational complexity. Can you provide a theoretical bound evaluating the computational complexity of solving the proposed semi-algebraic constraints as the state and action spaces scale beyond the two-dimensional toy models?
3. Do the isolated local maximizers observed in Figure 4 arise intrinsically from the fundamental structure of the POMDP, or are they an artifact of restricting the agent to the memoryless policy class?
4. How do standard policy optimization algorithms behave empirically within these complex geometric landscapes when applied to standard, higher-dimensional POMDP benchmarks?
5. What specific mathematical pathways exist to extend this semi-algebraic characterization to policies utilizing belief-state tracking?

**Limitations:**

The authors explicitly acknowledge that their results are structural, restrict the analysis to memoryless policies, and do not translate directly into efficient algorithms. However, they omit necessary technical discussion regarding the computational complexity of solving high-degree polynomial systems via quantifier elimination. Furthermore, the authors dismiss the potential societal impact of their work entirely. To improve the paper, the authors must incorporate a rigorous theoretical discussion on the computational scaling limits of the finite description, resolve the rank-deficiency singularities mathematically, expand the empirical evaluations to include standard POMDP benchmarks, and revise the impact statement to reflect the broader implications of foundational AI research.

**Strengths And Weaknesses:**

**Strengths And Weaknesses**
* **Originality (Strength):** The application of real algebraic geometry to parameterize the Bellman equation offers an original theoretical perspective. The authors successfully bridge the known polyhedral structure of MDPs and the complex, semi-algebraic nature of POMDPs.
* **Significance (Weakness):** The theoretical framework relies strictly on memoryless stochastic policies. To resolve state ambiguity, modern reinforcement learning practitioners routinely deploy history-dependent tracking, such as belief states, which effectively reduces the POMDP to a fully observable MDP. Therefore, it remains ambiguous whether the severe non-convexity and isolated local maximizers are intrinsic to the POMDP dynamics or merely artifacts of evaluating an inadequate policy class. Furthermore, the authors concede that these structural insights do not directly yield efficient algorithms for policy optimization, limiting immediate utility for the machine learning community.
* **Soundness (Weakness):** The derivation of the finite description in Section 4.1 masks a critical algebraic singularity. To convert geometric projections into polynomial equations, the authors compute the Moore-Penrose pseudoinverse, $D^+$. They state this explicit formulation holds specifically when the deviation matrix $D(x)$ exhibits full column or full row rank. However, the rank of $D(x)$ inherently depends on the variable $x$ itself. The manuscript omits necessary mathematical analysis of the ill-conditioned regions where $D(x)$ loses rank, rendering the finite description incomplete at these critical singularities. Additionally, the empirical validation utilizes trivial toy models restricted to two states and two actions. The manuscript completely lacks evaluations on standard POMDP benchmarks. This minimal empirical evidence fails to verify how the geometric complexity scales in high-dimensional state spaces or whether these isolated optima actually trap standard optimization algorithms in practice.
* **Presentation (Strength/Weakness):** The mathematical notation remains precise. The narrative transitions logically from infinite inequalities to finite descriptions. Figures 2, 3, and 4 effectively visualize the core geometric concepts in two dimensions. Conversely, the Impact Statement is abrupt and dismissive, asserting that no potential societal consequences "must be specifically highlighted here". This brevity fails to satisfy standard ethical disclosure guidelines for AI research.

---

> ### Author Rebuttal · Authors · 2026-03-31
>
> **1. Rank-deficiency of $D(x)$**
>
> The reviewer’s concern pertains to the use of the Moore-Penrose pseudoinverse in Section 4.1 as a means of deriving explicit equations for the condition $(I - DD^+)C = 0$. While this explicit representation depends on the rank of $D$, the underlying theoretical framework applies uniformly across all ranks. Indeed, when $D(x)$ drops rank, the expression for $D^+$ changes; however, the condition $(I - DD^+)C = 0$ remains equivalent to requiring that $C \in \operatorname{col}(D)$. This, in turn, is equivalent to the vanishing of all $(r+1)\times(r+1)$ minors of the augmented matrix $[D \mid C]$, where $r = \operatorname{rank}(D)$. Consequently, a semi-algebraic description can be obtained by stratifying according to the rank of $D$: for each fixed rank, the corresponding minors define a well-posed polynomial system, and the union over all ranks remains semi-algebraic by closure. We will include a detailed discussion of this rank stratification in the revision. We also note that Theorem 4.2 and Corollary 4.3 do not rely on the pseudoinverse.
>
> **2. Computational complexity of quantifier elimination**
>
> We constructed an explicit quantifier-free parametrization for the solution set and will be adding details in the revised appendix. While in general quantifier-free approaches can have a computational complexity which is doubly exponential in the number of variables, we show that we require only $k \choose l$ conditions, where $k$ is the product of the number of observations and number of actions and $l$ is the rank of the constraint matrix used to form the quantifier-free description.
>
> **3: Are isolated local optima intrinsic or artifacts?**
>
> The presence of isolated local optima is a consequence of geometric constraints induced by partial observability, rather than an artifact of the memoryless policy class itself. Indeed, policies with finite memory can be equivalently represented via observation enrichment: one augments the observation space with a memory component, so that memoryless policies in the augmented system correspond to policies with memory in the original system. Importantly, even after such augmentation, the system may remain partially observable, and the set of feasible value functions can still be subject to nonlinear constraints as described in our results, which in turn give rise to isolated local optima.
>
> We note that any limited-memory policy class can be recast as a class of Markovian policies through an appropriate enrichment of the observation space (see, e.g., Azizzadenesheli et al. ``Policy Gradient in Partially Observable Environments...'', 2020). Our framework does not rely on a specific structure or cardinality of the observation space; consequently, the geometric characterization we derive for memoryless POMDPs extends directly to such non-Markovian policy classes.
>
> We suggest that there is a natural trade-off: increasing memory can improve state identifiability, thereby reducing partial observability and potentially weakening the nonlinear constraints that shape the solution set. Quantifying this trade-off remains an interesting direction for future work, and our framework provides a natural starting point for such an analysis.
>
> **4: Behavior of standard algorithms on benchmarks**
>
> Prior work has observed that standard policy optimization methods can behave inconsistently in POMDP settings. Our results provide a geometric explanation: nonlinear constraints and isolated local optima can lead to sensitivity to initialization, variability across runs, and convergence to suboptimal solutions. We expect these effects to be amplified in higher-dimensional benchmarks. A systematic empirical study would be valuable, and our framework suggests directions such as examining convex relaxations and regularization to improve stability.
>
> **5: Extension to belief-state policies**
>
> For belief-state MDPs, the value function is defined over the belief state, which varies continuously. As shown by Sondik (1978), value functions corresponding to finitely transient policies are piecewise linear in the belief state; however, this does not characterize value functions in general. In our work, we characterize the set of feasible value functions for a given POMDP via a finite collection of polynomial equations and inequalities. Extending this perspective to belief-state policies would move beyond finite-dimensional polynomial descriptions to analytic expressions (polynomials in infinitely many variables). Nevertheless, it may still be possible to describe the finitely many linear regions in a semi-algebraic manner. Exploring this connection is an intriguing direction for future work.
>
> **6. Impact Statement**
>
> We used the standard ICML statement. Nonetheless, we appreciate the impetus to think more deeply here and will revise the statement to reflect a deeper consideration.

---

> > ### Author Rebuttal · Reviewer_eDuZ · 2026-04-01
> >
> > We thank the authors for their mathematically engaged rebuttal. We select option (c) because two core concerns remain unresolved and require revisions beyond a short rebuttal.
> >
> > Point 1 (Rank-deficiency): Resolved. The rank stratification argument is sound. Stratifying by rank and characterizing the column inclusion condition via minors of the augmented matrix [D|C] yields a well-posed polynomial system at each stratum, and the finite union preserves the semi-algebraic property. The clarification that Theorem 4.2 and Corollary 4.3 do not depend on the pseudoinverse further strengthens this resolution. We consider this concern addressed, provided the promised discussion appears in the revision.
> >
> > Point 2 (Computational complexity): Partially resolved. The (k choose l) bound is a concrete improvement over the general doubly exponential worst case. However, for large observation and action spaces, this binomial coefficient can itself grow rapidly. We look forward to the detailed derivation in the revised appendix and encourage the authors to include numerical estimates for representative problem sizes.
> >
> > Point 3 (Intrinsic vs. artifact): Partially resolved. The observation enrichment construction is valid: finite-memory policies can be recast as memoryless policies in an augmented observation space that remains partially observable. The paper's framework therefore extends in principle to non-Markovian policy classes. However, there is a gap between theoretical possibility and practical relevance. The authors acknowledge that increasing memory improves state identifiability and can weaken the nonlinear constraints giving rise to isolated local maximizers. Without quantifying this trade-off, we cannot determine whether the pathologies observed in the toy examples persist at practical memory sizes or dissolve as the policy class grows richer. This quantification requires new analysis that exceeds a rebuttal's scope.
> >
> > Point 4 (Standard benchmarks): Unresolved. The rebuttal provides no new empirical evidence. The manuscript evaluates only two-state, two-action models. Whether the isolated optima and non-convex structure identified in these minimal examples actually trap standard optimization algorithms in higher-dimensional POMDP benchmarks remains undemonstrated. The paper's value as a geometric explanation of optimization difficulty depends on establishing this empirical connection. Doing so requires new experiments.
> >
> > Point 5 (Belief-state extension): We appreciate the authors' candor that extending to belief-state policies moves beyond finite-dimensional polynomial descriptions. More broadly, the framework does not currently apply to the history-dependent policy classes (recurrent networks, transformers, finite observation windows) widely used in modern deep RL for partially observable settings. Combined with the unresolved trade-off in Point 3, this narrows the practical scope of the contribution.
> >
> > Point 6 (Impact Statement): Acknowledged.
> >
> > The rank-deficiency concern is resolved, and the complexity bound is a constructive step. However, the paper's core contribution is a geometric characterization of the POMDP optimization landscape. Two foundational questions remain: whether these geometric pathologies persist under richer policy classes (Point 3), and whether they manifest in realistic benchmarks (Point 4). Both require substantial new work. We encourage the authors to address these in a revision.

---

> > > ### Author Response · Authors · 2026-04-03
> > >
> > > ### Additional experiments with larger models
> > >
> > > To address the concern about empirical evidence, we conducted a systematic study across randomly generated POMDPs. For each configuration (S, A, O), we sampled independent instances and ran 50 random restarts of policy gradient over memoryless stochastic policies. The table below (abbreviated due to space constraints) reports for various configurations aggregate statistics over instances:
> > >
> > > (i) the spread of achieved values across restarts,
> > > (ii) the fraction of materially suboptimal runs (gap $> 0.01$), and
> > > (iii) the spread of the resulting policies.
> > >
> > > Across all configurations, we observe that **partial observability induces a substantial spread in both value and policy space**, whereas the fully observable baseline exhibits **negligible spread**.
> > >
> > > | Config (S,A,O) | Value Spread (Partial) | Value Spread (Full) | Subopt. Fraction (Partial) | Subopt. Fraction (Full) | Policy Spread (Partial) | Policy Spread (Full) |
> > > |---|---:|---:|---:|---:|---:|---:|
> > > | (8,2,2) | 26.695 ± 4.932 | 0.043 ± 0.051 | 0.220 ± 0.145 | 0.000 ± 0.000 | 0.314 ± 0.153 | 0.006 ± 0.008 |
> > > | (8,4,3) | 10.616 ± 7.909 | 0.204 ± 0.279 | 0.293 ± 0.220 | 0.050 ± 0.122 | 0.240 ± 0.148 | 0.054 ± 0.058 |
> > > | (12,2,2) | 15.348 ± 8.547 | 0.041 ± 0.037 | 0.197 ± 0.159 | 0.000 ± 0.000 | 0.279 ± 0.155 | 0.010 ± 0.019 |
> > > | (12,4,3) | 18.199 ± 6.600 | 0.151 ± 0.171 | 0.267 ± 0.149 | 0.000 ± 0.000 | 0.254 ± 0.119 | 0.019 ± 0.025 |
> > >
> > > To more directly quantify the **multi-optima structure**, we ran another experiment where for each initial state distribution $\rho$, we clustered the converged value functions. The number of clusters directly estimates the number of distinct local optima reached. The table below reveals that multiple distinct local optima are the norm rather than the exception. The fraction of (instance, $\rho$) pairs exhibiting multiple optima increases with $S$ as does the mean number of distinct local optima.
> > >
> > > | $S$ | Mean # optima (pooled over $O$) | Share Multi-optima (pooled over $O$) | Mean $\|V\|$ dist (pooled over $O$) |
> > > |---|---:|---:|---:|
> > > | 8  | 1.82 | 0.64 | 10.07 |
> > > | 12 | 2.05 | 0.75 |  9.78 |
> > > | 16 | 2.25 | 0.79 | 10.92 |
> > > | 20 | 2.67 | 0.88 | 11.53 |
> > > | 24 | 2.80 | 0.86 | 11.87 |
> > >
> > > These observations are consistent with our theoretical results stating that, under partial observability, the set of achievable value functions induced by memoryless policies forms a **semi-algebraic set with nontrivial geometry**, including curved boundaries, which can give rise to multiple local optima of different values. We will add these and more extensive experimental results (omitted here due to space constraints) in the revised paper along with scripts to reproduce them. We think these results offer excellent empirical evidence and substantively strengthen the manuscript, and thank the reviewer for helping us do so.
> > >
> > > ## Additional experiments with finite-memory policies
> > >
> > > To further investigate whether the observed optimization landscape is an artifact of restricting to memoryless policies, we extended our experiments to include **finite-memory policies** implemented via observation enhancement.
> > > The table below shows a consistent pattern across all tested configurations: **Increasing memory reduces the value spread** but **partial observability still induces significant spread**.
> > > This is consistent with the semi-algebraic characterization of the feasible value set. Increasing memory $k$ enlarges the policy class and set of feasible value functions, which can mitigate some optimization difficulties, reflected in the reduced spread. However, the geometric complexity induced by partial observability persists, as evidenced by remaining variability compared to the fully observable case.
> > >
> > > | Config (S,A,O) | Value Spread (partial_k0) | Subopt. Fraction (partial_k0) | Value Spread (partial_k1) | Subopt. Fraction (partial_k1) | Value Spread (partial_k2) | Subopt. Fraction (partial_k2) | Value Spread (full) | Subopt. Fraction (full) |
> > > |---|---:|---:|---:|---:|---:|---:|---:|---:|
> > > | (4,2,2) | 1.126 ± 2.407 | 0.315 ± 0.416 | 1.122 ± 2.410 | 0.110 ± 0.232 | 0.111 ± 0.239 | 0.125 ± 0.280 | 0.021 ± 0.022 | 0.000 ± 0.00 |
> > > | (4,3,2) | 1.339 ± 2.355 | 0.030 ± 0.054 | 0.243 ± 0.368 | 0.215 ± 0.427 | 0.292 ± 0.409 | 0.125 ± 0.240 | 0.055 ± 0.089 | 0.010 ± 0.022 |
> > > | (8,2,2) | 0.284 ± 0.385 | 0.205 ± 0.303 | 0.523 ± 0.719 | 0.260 ± 0.369 | 0.545 ± 0.819 | 0.310 ± 0.394 | 0.031 ± 0.016 | 0.000 ± 0.00 |
> > > | (8,3,2) | 0.287 ± 0.438 | 0.255 ± 0.423 | 0.305 ± 0.513 | 0.275 ± 0.418 | 0.138 ± 0.202 | 0.330 ± 0.401 | 0.113 ± 0.036 | 0.045 ± 0.087 |
> > >
> > > Although memory is not the main focus of the present article, we think its investigation is a highly interesting and concrete application of our framework, and we will continue to extend these experiments, and also include an appendix in the revision using our explicit quantifier free descriptions to explicitly compare the degree of the boundary components depending on $k$.

---

### Decision · Program_Chairs · 2026-04-30

**Decision:**

Accept (regular)

**Comment:**

This paper gives a novel geometric characterization of feasible value functions in infinite-horizon POMDPs under memoryless stochastic policies, showing that the feasible set is semi-algebraic rather than polyhedral as in fully observable MDPs. Reviewers agreed that the contribution is original and mathematically interesting. The main concerns were about handling rank-deficient cases, computational relevance, the restriction to memoryless policies, and the initially limited empirical support. The authors addressed these points constructively in rebuttal, including clarifying the rank-stratification argument, explaining the MDP special case, and adding experiments on larger random POMDPs and finite-memory variants. While the paper remains limited in scope and its practical algorithmic implications are still somewhat indirect, the theory appears sound and the overall reviewer consensus after rebuttal is positive. I therefore recommend accept.

For the final version, the authors should revise the paper accordingly by:
(i) incorporating the rebuttal clarifications on rank-deficient cases and the MDP special case;
(ii) clearly adding the new empirical results referenced in the discussion;
(iii) strengthening the discussion of scope and limitations, especially regarding richer policy classes; and
(iv) improving the explanation of how the semi-algebraic characterization connects to optimization behavior in practice.